# Revealing the atomic and electronic mechanism of human manganese superoxide dismutase product inhibition

Jahaun Azadmanesh [1], Katelyn Slobodnik [1], Lucas R. Struble[1], William E. Lutz[1], Leighton Coates [2], Kevin L. Weiss [3], Dean A. A. Myles [3], Thomas Kroll[4] & Gloria E. O. Borgstahl [1] ✉

Human manganese superoxide dismutase (MnSOD) is a crucial oxidoreductase that maintains the vitality of mitochondria by converting superoxide ($O_2^{\bullet-}$) to molecular oxygen ($O_2$) and hydrogen peroxide ($H_2O_2$) with proton-coupled electron transfers (PCETs). Human MnSOD has evolved to be highly product inhibited to limit the formation of $H_2O_2$, a freely diffusible oxidant and signaling molecule. The product-inhibited complex is thought to be composed of a peroxide ($O_2^{2-}$) or hydroperoxide ($HO_2^-$) species bound to Mn ion and formed from an unknown PCET mechanism. PCET mechanisms of proteins are typically not known due to difficulties in detecting the protonation states of specific residues that coincide with the electronic state of the redox center. To shed light on the mechanism, we combine neutron diffraction and X-ray absorption spectroscopy of the product-bound, trivalent, and divalent states of the enzyme to reveal the positions of all the atoms, including hydrogen, and the electronic configuration of the metal ion. The data identifies the product-inhibited complex, and a PCET mechanism of inhibition is constructed.

Electron transfers are catalyzed by oxidoreductases that make up about a quarter of known enzymes[1]. Essential biological processes such as energy generation and DNA synthesis rely on oxidoreductases, and dysfunction results in a wide range of diseases[2–6]. To accelerate the electron transfers to rates needed for life, oxidoreductases couple the transfer of electrons to the transfer of protons in a process called proton-coupled electron transfer (PCET)[1,7,8]. Defining mechanisms of PCETs in enzymes is challenging as it requires precise definition of proton donors/acceptors in tandem with electron transfer steps[7,9,10]. Defining these fundamental biochemical reactions is important for understanding disease and designing therapeutic and industrial applications[11–14].

Through PCETs, oxidoreductases regulate the concentration of reactive oxygen species (ROS) in cells. Dysregulation of ROS concentrations contribute to cardiovascular disease, neurological disease, and cancer progression[15,16]. In the mitochondrial matrix, the oxidoreductase human manganese superoxide dismutase (MnSOD) decreases $O_2^{\bullet-}$ concentrations using PCETs. ($k_{cat}/K_m > \sim 10^9 \, M^{-1} s^{-1}$)[17,18]. MnSOD eliminates $O_2^{\bullet-}$ by oxidation to $O_2$ with a trivalent Mn ion ($k_1$) and reduction to $H_2O_2$ with a divalent Mn ion ($k_2$) in two half-reactions that restore the trivalent Mn ion[17,19].

$$Mn^{3+} + O_2^{\bullet-} \leftrightarrow Mn^{2+} + O_2 \qquad k_1 = 1.5 \, nM^{-1} s^{-1}$$
$$Mn^{2+} + O_2^{\bullet-} + 2H^+ \leftrightarrow Mn^{3+} + H_2O_2 \quad k_2 = 1.1 \, nM^{-1} s^{-1}$$

Endogenous $O_2^{\bullet-}$ is produced from electrons leaking from the mitochondrial electron transport chain[20,21]. MnSOD is the only enzyme that the mitochondrial matrix has to lower $O_2^{\bullet-}$ levels to avoid

[1]Eppley Institute for Research in Cancer and Allied Diseases, 986805 Nebraska Medical Center, Omaha, NE 68198-6805, USA. [2]Second Target Station, Oak Ridge National Laboratory, 1 Bethel Valley Road, Oak Ridge, TN 37831, USA. [3]Neutron Scattering Division, Oak Ridge National Laboratory, 1 Bethel Valley Road, Oak Ridge, TN 37831, USA. [4]Stanford Synchrotron Radiation Lightsource, SLAC National Accelerator Laboratory, Menlo Park, CA 94025, USA. ✉e-mail: gborgstahl@unmc.edu

macromolecular damage[22]. Therefore, abnormalities in MnSOD activity compromise mitochondrial function and lead to disease. Indeed, genetic aberrations of MnSOD are associated with several cancer types, with breast and prostate cancers being the most frequent in UniProt and ClinGen databases[23,24]. Polymorphisms of MnSOD are a predictor for deficient cardiovascular function[25]. These disease states indicate that MnSOD's PCETs are key to preserving health.

MnSOD relays protons to the active site for PCET catalysis through its hydrogen bond network (dashed blue lines, Fig. 1a). The Mn ion is bound by inner-sphere residues His26, His74, His163, Asp159, and a solvent molecule denoted as WAT1. The hydrogen bond network extends from WAT1 to outer-sphere residues Gln143, Tyr34, another solvent molecule denoted as WAT2, His30, and Tyr166 from the neighboring subunit. The space between His30 and Tyr34 is the only opening available for solvent and substrate/product to enter or exit the active site. Hydrophobic residues Trp123, Trp161, and Phe66 hold the hydrogen bonding atoms of Asp159, WAT1, Gln143, and Tyr34 close together[26]. Our previous work detailed changes in protonations and hydrogen bonding between Mn³⁺SOD and Mn²⁺SOD, giving insight into how the active site metal ion redox state alters the $pK_a$s of nearby amino acids[27].

Neutron crystal structures of wildtype MnSOD at different oxidation states revealed several changes in protonation and hydrogen bonding[27]. Studying metalloenzyme PCET mechanisms with neutron crystallography is advantageous because metal oxidation states are unaltered (unlike X-rays), and the neutron scattering of deuterium is similar to carbon, nitrogen, and oxygen[28,29]. These neutron structures gave three important observations about the MnSOD mechanism. First, an unusual proton transfer occurred between WAT1 and Gln143 during the Mn³⁺ → Mn²⁺ transition. Gln143 transferred an amide proton to a metal-bound ⁻OH molecule, WAT1, to form an amide anion and $H_2O$ (Fig. 1b, c). Two short-strong hydrogen bonds (SSHBs) of Gln143 with WAT1 and Trp123 stabilize the amide anion. A SSHB is a type of hydrogen bond that stabilizes catalytic steps and enhances kinetic rates (hashed lines, Fig. 1c)[30–32]. SSHBs are defined as hydrogen bonds that are less than 1.8 Å between the donor D/H atom and the acceptor atom. Second, Tyr34 is deprotonated in the Mn³⁺ redox state and becomes protonated in the Mn²⁺ state (Fig. 1d, e). Tyr34 probably provides one of the two protons needed during the second half-reaction ($k_2$), where $H_2O_2$ forms from the protonation of substrate[18,33–35]. Third, proton transfers occurred between Tyr166 and Nᵉ² of His30 employing a low-barrier hydrogen bond (LBHB) that

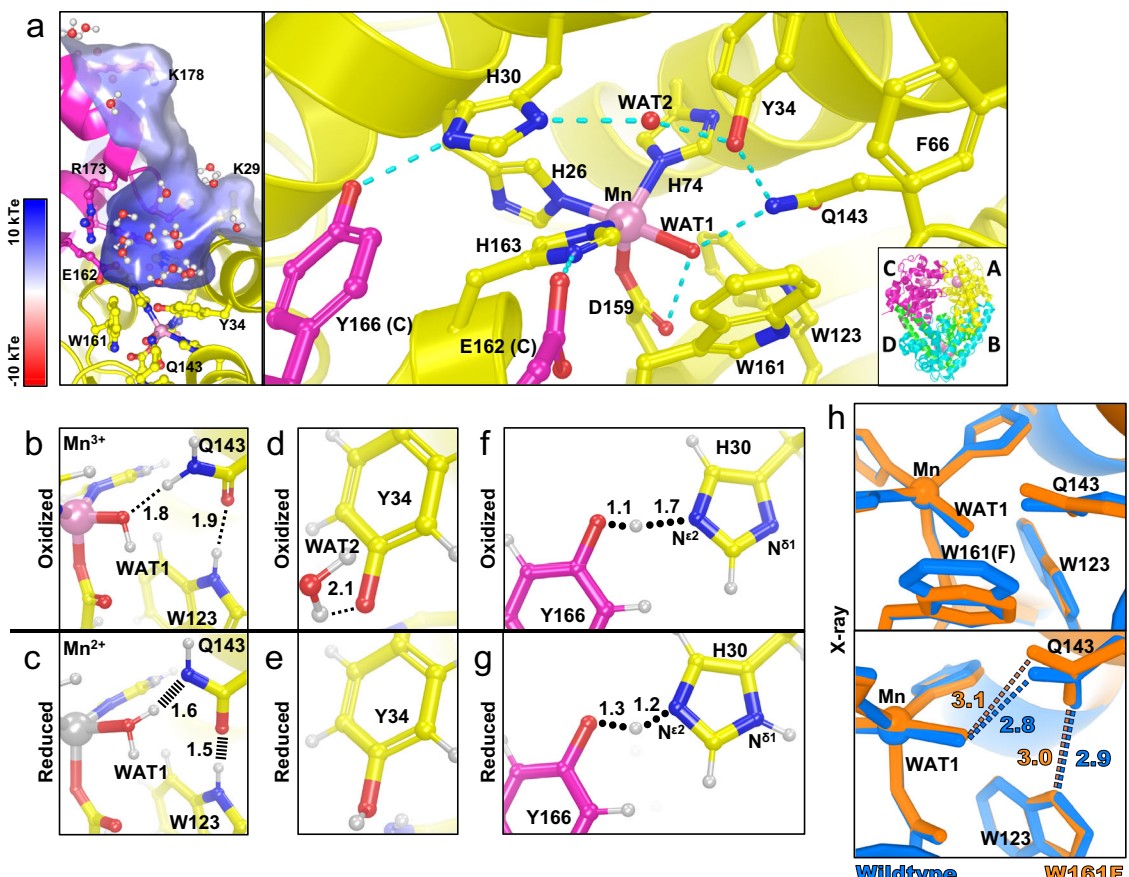

**Fig. 1 | Structure of human wildtype MnSOD and protonation changes coupled to oxidation states. a** The active sites of tetrameric MnSOD are in a positively charged cavity between two subunits. Blue dashes indicate hydrogen bonds. The inset indicates chain identity where the dimeric crystallographic asymmetric unit is composed of chains A and B while C and D are generated by symmetry. **b, c** Room temperature neutron structures of Mn³⁺SOD and Mn²⁺SOD show changes in protonation and hydrogen bond distances for WAT1, Gln143, and Trp123. Dotted lines indicate hydrogen bonding and hashed lines indicate SSHBs. **d, e** Tyr34 is observed deprotonated in Mn³⁺SOD and protonated in Mn²⁺SOD. **f, g** A shared proton is seen between His30 and Tyr166, indicated by round dots. The Nᵟ¹ of His30 is deprotonated when the Mn ion is oxidized and protonated when the Mn ion is reduced.

**a** was created from MnSOD X-ray structure (PDB ID 5VF9)[78], **b**, **d**, and **f** are from the Mn³⁺SOD neutron structure (PDB ID 7KKS)[27], and **c**, **e**, and **g** are from the Mn²⁺SOD neutron structure (PDB ID 7KKW)[27]. All hydrogen positions were experimentally determined with the exception of solvent molecules in (**a**) that were randomly generated to accentuate the solvent in the active site funnel. **h** Active site overlay of wildtype MnSOD (blue, PDB ID 5VF9 solved at 1.82 Å resolution)[78] and Trp161Phe MnSOD (orange, PDB ID 8VJ8 solved at 1.70 Å resolution, this work) displaying distortion in the WAT1-Gln143 interaction. Hydrogen bond distances for the neutron structures (**b–g**) are between the donor D atom and acceptor atom and are denoted in Å. For X-ray structures (**h**), hydrogen bond distances are between oxygen and nitrogen.

**Table 1 | Individual steady-state rate constants of MnSOD[a]**

| MnSOD[a] | $k_1$ (nM$^{-1}$ s$^{-1}$) | $k_2$ (nM$^{-1}$ s$^{-1}$) | $k_3$ (nM$^{-1}$ s$^{-1}$) | $k_4$ (s$^{-1}$) |
|---|---|---|---|---|
| Human Wildtype | 1.5 | 1.1 | 1.1 | 117 |
| Human W161F | 0.3 | <0.01 | 0.46 | 33 |

[a]pH 8.2 and 25 °C from Hearn et al.[26].

coincided with changes in the protonation of N$^{\delta 1}$ of His30 (Fig. 1f, g). A LBHB is a type of hydrogen bond where the heteroatoms transiently share a proton, and the hydrogen bond distances between the donor D/H atom and the heteroatoms are nearly equivalent (1.2–1.3 Å)[36]. Altogether, this study revealed that electron transfers are coupled with several proton transfer events, and the proximity of the active site metal ion creates several unusual amino acid pK$_a$s that are essential to the PCET mechanism.

Whether MnSOD uses an inner- or outer-sphere mechanism for catalysis is unclear[18,35,37]. For an inner-sphere mechanism, $O_2^{\bullet-}$ may bind either the open sixth-coordinate position (opposite Asp159, Fig. 1a) or displace the metal-bound solvent (WAT1, Fig. 1a)[18,34,35,37]. In an outer-sphere mechanism, $O_2^{\bullet-}$ hydrogen bonds between His30 and Tyr34 for PCET. Indeed, quantum chemistry calculations have postulated that the first half-reaction ($k_1$) proceeds through an inner-sphere mechanism while the second half-reaction ($k_2$) uses an outer-sphere mechanism[34]. The outer-sphere mechanism ascribed to the second half-reaction ($k_2$) is compelling because His30 or Tyr34 could serve as proton donors for the formation of $H_2O_2$. Regardless, definitive experimental evidence for how $O_2^{\bullet-}$ interacts with the active site is lacking.

MnSOD is highly inhibited by its product, $H_2O_2$, a freely diffusible oxidant and signaling molecule. Notable roles of $H_2O_2$ generated from MnSOD include stimulating apoptotic signaling pathways[38,39], coordinating protein localization and activity[40], mitochondrial biogenesis[41], and inactivating protein tyrosine phosphatases like the tumor suppressor phosphatase and tension homolog (PTEN)[42]. These roles of mitochondrial-derived $H_2O_2$ may explain the evolutionary adaptation of human MnSOD to be more product-inhibited compared to its prokaryotic counterparts[35,37]. Human abnormalities in $H_2O_2$ steady-state concentrations are hallmarks of disease[43,44], and expression of MnSOD mutants that have deficient product inhibition leads to elevated cellular ROS levels compared to wildtype enzyme[45]. Due to the multifaceted roles of $H_2O_2$ in cellular physiology, MnSOD product inhibition ensures that $H_2O_2$ concentrations are tightly regulated in conjunction with other $H_2O_2$ scavengers.

Product inhibition is dependent on the relative concentrations of $O_2^{\bullet-}$ to MnSOD. Past experiments show that at low ratios of $[O_2^{\bullet-}]$:[MnSOD], catalysis proceeds through the fast $k_1$ and $k_2$ half-reactions[35,46]. When the concentration of $O_2^{\bullet-}$ was much higher than MnSOD, $O_2^{\bullet-}$ dismutation decreased with an initial fast step (first order) and then a slower step (zero-order). The kinetics were attributed to a branching mechanism starting from $Mn^{2+}$SOD, where $O_2^{\bullet-}$ either rapidly converts to $H_2O_2$ ($k_2$, first-order) or $Mn^{2+}$SOD forms an inhibited complex ($k_3$, first-order) that disassociates with a protonation event ($k_4$, zero-order)[35,37]. While the chemical equations of $k_3$ and $k_4$ added together resemble that of $k_2$, the zero-order process of $k_4$ means that $O_2^{\bullet-}$ is not reacting with MnSOD during this phase, and the enzyme is product-inhibited. The MnSOD $k_2$:$k_3$ gating ratio defines whether catalysis proceeds through the fast pathway to form $Mn^{3+}$ and $H_2O_2$ ($k_2$) or the slow zero-order pathway ($k_3$ and $k_4$). Human MnSOD has the lowest known gating ratio, with $k_2$ equal to $k_3$, and thus the highest propensity to be product-inhibited (Table 1)[35,37,47]. This roughly means that at $O_2^{\bullet-}$ concentrations that are much higher than MnSOD, 50% of reactions with $Mn^{2+}$SOD form the inhibited complex. Mechanistically, product inhibition is thought to be initiated from $Mn^{2+}$ and $O_2^{\bullet-}$ to form a complex presumed to be $[Mn^{3+}\text{-}O_2^{2-}]$ or $[Mn^{3+}\text{-}^-OOH]$

($k_3$). Relief of inhibition occurs with at least one protonation to produce $Mn^{3+}$ and $H_2O_2$ ($k_4$)[35,37]. However, the oxidation and protonation states of the inhibited complex (depicted in square parenthesis above and in $k_3$ and $k_4$ below) have yet to be experimentally determined[35,37]. Our work seeks to define the product-inhibited complex.

$$Mn^{2+} + O_2^{\bullet-} \leftrightarrow \left[ Mn^{3+}\text{-}O_2^{2-} \right] \qquad k_3 = 1.1\,nM^{-1}s^{-1}$$

$$\left[ Mn^{3+}\text{-}O_2^{2-} \right] + 2H^+ \leftrightarrow Mn^{3+} + H_2O_2 \qquad k_4 = 120\,s^{-1}$$

or

$$Mn^{2+} + O_2^{\bullet-} + H^+ \leftrightarrow \left[ Mn^{3+}\text{-}^-OOH \right] \qquad k_3 = 1.1\,nM^{-1}s^{-1}$$

$$\left[ Mn^{3+}\text{-}^-OOH \right] + H^+ \leftrightarrow Mn^{3+} + H_2O_2 \qquad k_4 = 120\,s^{-1}$$

Current mechanistic insight into human MnSOD product inhibition has relied on kinetic studies of point mutants that alter the competing reaction pathway steps of $k_2$ and $k_3$. Of particular interest is the Trp161Phe variant that decreases $k_2$ by more than two orders of magnitude compared to wildtype, leading to exclusive use of the product-inhibited pathway as $k_2 \ll k_3$ (Table 1)[26,37,48]. This roughly means that when $O_2^{\bullet-}$ levels are much greater than Trp161Phe MnSOD, ~99% of reactions with Trp161Phe $Mn^{2+}$SOD form the inhibited complex. In addition, Trp161Phe MnSOD's $k_4$ is decreased four-fold. This means that the product-inhibited complex persists longer than the wildtype. The Trp161Phe variant also preserves the hydrogen bond network involved in catalytic PCETs. For these reasons, Trp161Phe MnSOD was selected for experiments to trap and reveal the product-inhibited state.

The Trp161Phe variant slightly distorts the nearby positions of WAT1 and Gln143, suggesting that they are key factors in product inhibition (Fig. 1h)[26,49]. Our previous work demonstrated that tight hydrogen bonding between Gln143, WAT1, and Trp123 is present in $Mn^{2+}$SOD (Fig. 1b) and may be required for fast PCET catalysis through the uninhibited pathway (i.e., reaction $k_2$)[27]. Trp161 is immediately adjacent to the WAT1 position and sterically holds the solvent molecule in the correct orientation for interaction with Gln143. The reduction of amino acid size at position 161 from tryptophan to phenylalanine leads to a slight lengthening of the distance between WAT1 and Gln143 (Fig. 1h). This longer hydrogen bond distance between WAT1 and Gln143 may be related to the near ablation of the $k_2$ redox reaction (Table 1). Other enzymes that rely on proton transfers for enzymatic activity, such as α-chymotrypsin, substantially increase kinetic rates with stronger hydrogen bonds[31]. Another possibility is that WAT1 in the Trp161Phe variant can be more easily displaced from the Mn ion by substrate or product binding. This would end up blocking the back-and-forth proton transfers between Gln143 and WAT1 we observed during redox cycling of the Mn ion[27]. Thus, the strategic Trp161Phe variant is advantageous for studying MnSOD catalysis and allows us to determine the significance of the Gln143-WAT1 interaction in product inhibition.

Interestingly, some reports have observed that high concentrations of $H_2O_2$ with $Mn^{3+}$SOD lead to the product-inhibited complex that decays to $Mn^{2+}$SOD[50]. $H_2O_2$ acting as a reducing agent for the enzyme is unusual and suggests that a back reaction is involved that produces $O_2^{\bullet-}$ and $Mn^{2+}$SOD[26,50–52]. Mutagenesis studies found that the formation of the product-inhibited complex with excesses of $H_2O_2$ to be dependent on the $k_2$:$k_3$ ratio, which implicates the involvement of the forward reactions[50]. This potentially means that the $O_2^{\bullet-}$ and $Mn^{2+}$SOD produced from the back reaction could then react in the forward direction to form the inhibited complex. Isolating the inhibited complex in this manner would circumvent the use of aqueous $O_2^{\bullet-}$ that has a half-life of 10 s or less depending on pH and concentration[37,53,54]. For

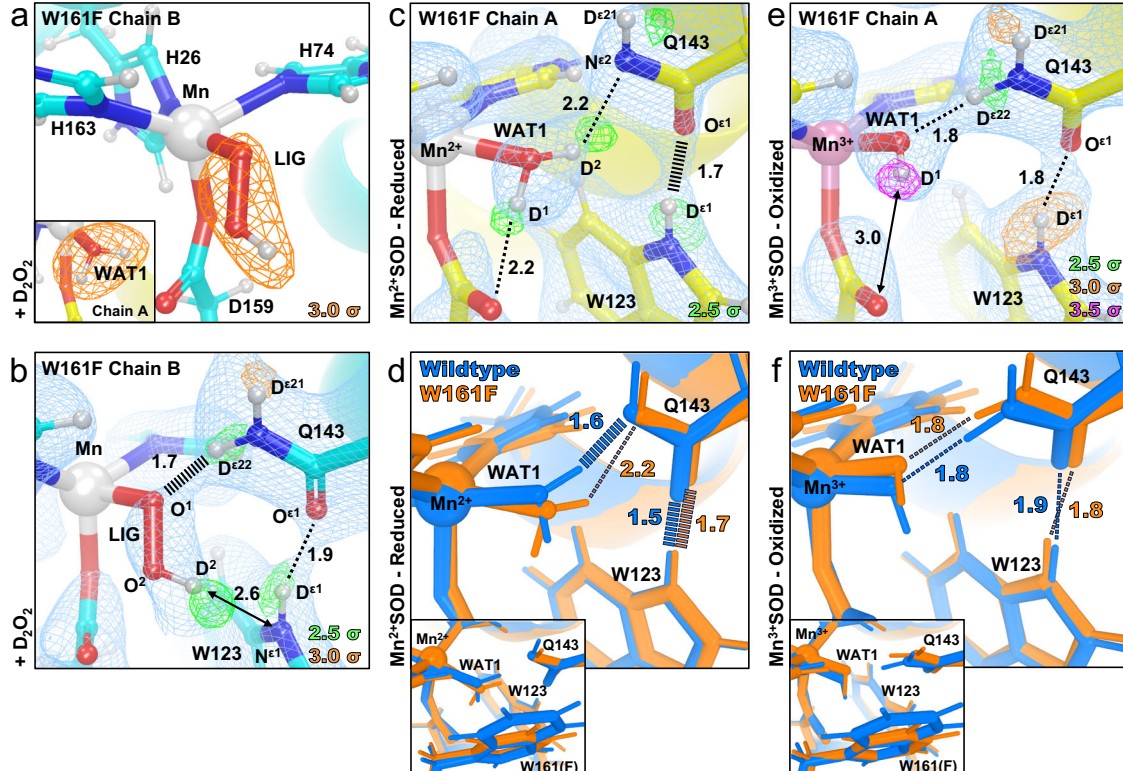

**Fig. 2 | Neutron structures and protonation states at the active site of $D_2O_2$-soaked, reduced, and oxidized Trp161Phe MnSOD. a** $D_2O_2$-soaked Trp161Phe MnSOD for chain *B* with a singly-protonated dioxygen ligand (denoted LIG). Chain A is displayed as an inset to highlight differences in the difference density shapes that led to the identification of LIG in chain *B*. **b** $D_2O_2$-soaked Trp161Phe MnSOD for chain *B*. **c** Trp161Phe $Mn^{2+}$SOD for chain *A*. **d** Active site overlay of wildtype $Mn^{2+}$SOD and Trp161Phe $Mn^{2+}$SOD demonstrating differences in hydrogen bond strength of the Gln143 amide anion and movement of WAT1. Inset highlights the position of the residue 161 mutation. **e** Trp161Phe $Mn^{3+}$SOD for chain *A*. **f** Active site overlay of wildtype $Mn^{3+}$SOD and Trp161Phe $Mn^{3+}$SOD. Inset highlights the position of the residue 161 mutation. Green, orange, and magenta omit $|F_o| - |F_c|$ difference neutron scattering length density of protons displayed at 2.5σ, 3.0σ, and 3.5σ, respectively. Light blue $2|F_o| - |F_c|$ density is displayed at 1.0σ. Distances are in Å. Dashed lines indicate typical hydrogen bonds and hashed lines indicate SSHBs that are hydrogen bonds <1.8 Å. All neutron structures were solved to 2.3 Å resolution, including those of wildtype MnSOD from our previous work[27]. For the resting state structures, only one active site is shown due to high structural similarities; see Supplementary Fig. 2 for the other active site. All neutron structures were solved to 2.3 Å resolution.

these reasons, we used $H_2O_2$ soaking in combination with the Trp161Phe MnSOD variant to characterize the inhibited complex.

Here, we sought to define the mechanism of human MnSOD product inhibition by (1) identifying how the product binds to the active site, (2) discerning the protonation states of the product-inhibited complex, and (3) determining the electronic configuration of the complex. To pursue these goals, we used neutron crystallography to determine the position of every atom (including H/D), X-ray absorption spectroscopy (XAS) to identify the geometric and electronic structure of the metal ion and its ligands, and quantum mechanical (QM) chemistry calculations to help interpret the experimental data. For oxidoreductases, the neutron structures are noteworthy because this is one of the first visualizations of a redox center-dioxygen species interaction without radiation-induced artefacts[55].

## Results and discussion
### The structural identity of the product-inhibited complex
To visualize the atomic identity of the product-inhibited complex and the corresponding active site protonation states, a cryocooled neutron structure of perdeuterated Trp161Phe MnSOD soaked with deuterium peroxide ($D_2O_2$) was solved at 2.30 Å resolution. We pursued the neutron structure to visualize Mn-ligand interactions without radiation-induced effects[29]. The crystallization at pH 7.8 allowed the identification of all the proton positions in the active site at the physiological pH of mitochondria. Enzymes that use proton transfers to facilitate catalysis, like MnSOD, are expected to undergo changes in

protonation states depending on the presence of substrate or product[56-58]. We chose the Trp161Phe variant to study product inhibition for its propensity to accumulate and retain the product-inhibited complex[26,37,50,51,59]. First, we solved the all-atom structure of the entire enzyme excluding the active site. Then, with these phases, we carefully interrogated the active site nuclear density maps for visually distinct peroxide-derived species and protons on O or N atoms.

The first $|F_o| - |F_c|$ nuclear scattering length density we examined immediately showed that the two subunits of the crystallographic asymmetric unit contained different species bound to the manganese ion, opposite His26 (Fig. 2a). Differences between the two subunits are often observed for the MnSOD $P6_122$ crystal form because their solvent accessibility is not the same[27]. The density of chain *A* has the characteristic oval shape previously seen in reduced wildtype MnSOD corresponding to the metal-bound water molecule WAT1[27]. For chain *B*, the density is more elongated and in an orientation nearly parallel with the Mn-Asp159 bond. We interpreted this density as a dioxygen species that has displaced WAT1 to bind the Mn ion (denoted as LIG for ligand, Fig. 2a). LIG refined well with full occupancy. The corresponding X-ray structures (Supplementary Fig. 1) and a previous peroxide-soaked *Escherichia coli* wildtype MnSOD structure support this interpretation of the nuclear density[59,60]. The dioxygen molecules in these X-ray structures are different in orientation and partially occupied due to the effects of X-ray irradiation. They demonstrate the benefit of using neutrons for radiation-free structural analysis. At physiological temperatures, thermochromism spectra also suggest a displacement

of WAT1 for binding of a dioxygen species in agreement with our cryocooled crystallographic data[61]. From the nuclear density, we conclude that a dioxygen species displaced the metal-bound solvent molecule to form a five-coordinate complex.

Next, the protonation states near the Mn ion and the dioxygen ligand that correspond to product inhibition were examined. The dioxygen species has a single proton (labeled as $D^2$, Fig. 2b) on $O^2$ and points toward the $N^{\varepsilon 1}$ of Trp123, but not at a hydrogen bonding distance. Given that the crystal was soaked with $D_2O_2$ before data collection, it can be presumed that the molecule was deprotonated first and then displaced WAT1 to bind the Mn ion. For Gln143, rather than being deprotonated to an amide anion that as seen in wildtype $Mn^{2+}SOD$[27] and Trp161Phe $Mn^{2+}SOD$ (Fig. 2c, d), a neutral amide is present. $D^{\varepsilon 22}$(Gln143) and $O^1$(LIG) form a 1.7 Å SSHB. This SSHB may be a possible conduit for proton transfer to occur, especially since a similar proton transfer was directly observed in wildtype MnSOD (Fig. 1b, c). The product-inhibited complex consists of a singly-protonated dioxygen ligand forming a SSHB with neutral Gln143.

The coordination number of the Mn ion for product inhibition has long been debated[35,37,52,62–64]. One proposed mechanism from QM calculations was that a dioxygen species binds to $Mn^{3+}$ at the sixth ligand position opposite Asp159[61–63]. An elegant study from Sheng and colleagues on *Saccharomyces cerevisiae* MnSOD observed the formation of a six-coordinate $Mn^{3+}$ ion complex from electron paramagnetic resonance (EPR) and supported the possibility that $Mn^{3+}$ may become six-coordinated during inhibition[47]. A second proposal was that a dioxygen species replaces the position of the solvent ligand as seen in optical absorption spectra and in an $H_2O_2$-soaked wildtype *E. coli* MnSOD X-ray crystal structure[60,61]. However, such a five-coordinate inhibited complex in the *E. coli* structure has been interpreted to be less likely as the dioxygen species was partially occupied and introduced steric crowding that led to higher computational energies[37,62,63]. X-ray structures are further complicated by radiation effects that lead to bond distances that are not representative of catalytic conditions[29]. Our radiation-free neutron diffraction suggests that product inhibition in human MnSOD is a five-coordinate complex with a singly-protonated dioxygen ligand displacing the metal-bound solvent molecule. Although sterically crowded, this complex prohibits catalysis during its lifetime by blocking the fast back-and-forth proton transfer between WAT1 and Gln143 during PCET redox cycling (Fig. 1b, c)[27]. We explore the valency of the Mn ion and the identity of the dioxygen species in later sections.

## The Gln143-WAT1 SSHB of Mn$^{2+}$SOD limits product inhibition

Next, we look for clues as to what structural features lead to product inhibition, why the Trp161Phe variant is highly product-inhibited, and why Trp161 is needed for catalysis. To begin to understand the effects of the point mutation, we obtained a cryocooled 2.30 Å resolution neutron structure of Trp161Phe $Mn^{2+}SOD$ where the metal ion was chemically reduced with dithionite (Fig. 2c). Dithionite reduces the Mn ion without entering the active site[27,65]. The Mn bond distances of five-coordinate Trp161Phe $Mn^{2+}SOD$ are similar to wildtype $Mn^{2+}SOD$ (Supplementary Table 1). Moreover, the protonation states resemble that of wildtype $Mn^{2+}SOD$, the bound solvent molecule is $D_2O$ (not $^-OD$) and Gln143 is deprotonated to an amide anion (Figs. 1c, 2d). Deprotonated amino acids are confirmed when attempts to model and refine a proton result in negative $|F_o| - |F_c|$ difference neutron scattering length density and all the other protons of the amino acid can be placed. The Trp161Phe variant and wildtype contrast at the Gln143-WAT1 interaction. In wildtype, the Gln143 amide anion forms a 1.6 Å SSHB with WAT1 (Fig. 2d)[27]. For the variant, there is a weaker 2.2 Å hydrogen bond between Gln143 and WAT1. These observations suggest that for the reduced enzyme, the role of Trp161 is to hold Gln143 and WAT1 in a tight hydrogen bonding interaction.

We next measured a room temperature Trp161Phe $Mn^{3+}SOD$ neutron structure at 2.30 Å resolution (Fig. 2e). Cryocooling and chemical treatment were not pursued due to a limited supply of perdeuterated crystals during beamtime. However, purified human MnSOD is ~90% oxidized[66] and the resultant Trp161Phe Mn-ligand bond distances strongly resemble that of the chemically oxidized wildtype $Mn^{3+}$ counterpart (Supplementary Table 2). Furthermore, the hydrogen bonding and protonation states between variant and wildtype are identical, where WAT1 is a $^-OD$ molecule and Gln143 is of the neutral amide form (Fig. 2f). Due to the high similarity in structure between Trp161Phe and wildtype in the oxidized state, we conclude that the catalytic consequence of the variant for the $Mn^{3+}$ to $Mn^{2+}$ redox transition is not related to the interactions between the molecules of WAT1, Gln143, and Trp123. Instead, catalytic differences may be related to subtle electronic differences or alterations elsewhere in the active site (*vide infra*).

The Trp161Phe variant has historically been puzzling as it profoundly affects PCET catalysis without directly removing a hydrogen bond donor or acceptor[26,48,51]. The fast redox transition of $Mn^{2+}$ to $Mn^{3+}$ is nearly ablated ($k_2$, Table 1) while the formation of the inhibited complex is enriched ($k_2 \ll k_3$, Table 1) with a slow disassociation ($k_4$, Table 1)[26,37,48]. From the cryocooled neutron structure of Trp161Phe $Mn^{2+}SOD$ (Fig. 2c), it is apparent that the Gln143-WAT1 hydrogen bonding interaction is perturbed compared to wildtype $Mn^{2+}SOD$ (Fig. 2d) and may be the primary cause of the observed kinetic effects. The bulky Trp161 residue likely constrains Gln143 and WAT1 to form a SSHB for proton transfers. We previously established that a strong $O(WAT1)-D^2(WAT1)-N^{\varepsilon 2}(Gln143)$ interaction is needed during the $Mn^{2+}SOD$ oxidation state for a proton transfer to occur during the $Mn^{2+}$ to $Mn^{3+}$ PCET[27]. Mutation of glutamine 143 to asparagine similarly ablates catalysis for the $Mn^{2+}$ to $Mn^{3+}$ half-reaction, and a longer WAT1-Gln interaction distance among isoforms of MnSODs correlate with decreased catalytic rates[35,67]. In summary, the structural features that lead to product inhibition include (1) the displacement of WAT1 by a dioxygen species (Figs. 2a), (2) when WAT1 is easier to displace product inhibition is increased (Fig. 2d, Table 1), and (3) a major role for Trp161 is to promote a strong Gln143-WAT1 interaction for proton transfer.

## The electronic identity of the product-inhibited complex

The coordination of dioxygen species and the oxidation state of the Mn ion for product inhibition has long been under debate[35,37,52,62–64]. Therefore, we pursued Mn K-edge XAS of MnSOD to cross-validate our crystallographic data and inform our refined structure. K-edge XAS can be divided into two regions: (1) the extended X-ray absorption fine structure (EXAFS) region at higher energies that contains structural information and (2) the X-ray absorption near edge structure (XANES) region that is reflective of the metal oxidation and geometric state. EXAFS of the peroxide-soaked Trp161Phe MnSOD complex was sought to obtain Mn covalent bond distance information for comparison to the neutron structure and QM density functional theory (DFT) calculations (Table 2). XANES was similarly pursued to identify the oxidation state and cross-validate the Mn ion's geometry corresponding to the inhibited complex. X-ray irradiation was carefully monitored so that two subsequent scans of the same spot did not have photoreduction differences, and different positions on the sample were used for each scan.

We first address the EXAFS structure of peroxide-soaked Trp161Phe MnSOD. The Fourier transform of the raw EXAFS spectra yields the atomic radial distribution around the absorbing Mn ion and is referred to as *R* space (Fig. 3a). The EXAFS spectrum exhibits a peak centered at ~2.1 Å that is best fit by two sets of scatterers, three nitrogen atoms at 2.20 Å and two oxygen atoms at 2.04 Å (Supplementary Table 2). These distances agree with those of Mn ion bond lengths seen in the neutron structure and DFT geometry optimizations using a

**Table 2 | Comparison of peroxide-soaked Trp161Phe MnSOD bond lengths from EXAFS fits, neutron structures, and DFT calculations[a]**

| Bond | EXAFS (Å) | Neutron Structure (Å)[c] | XANES Fit (Å) | DFT (Å)[d] [Mn²⁺-⁻OOH] | DFT (Å)[e] [Mn²⁺-·OOH] | DFT (Å)[f] [Mn³⁺-⁻OOH] |
|---|---|---|---|---|---|---|
| Mn-N$^{\epsilon 2}$(H26) | 2.20 | 2.20 | 2.13 | 2.20 | 2.12 | 2.02 |
| Mn-N$^{\epsilon 2}$(H74) | 2.20 | 2.28 | 2.22 | 2.21 | 2.15 | 2.09 |
| Mn-N$^{\epsilon 2}$(H163) | 2.20 | 2.17 | 2.16 | 2.20 | 2.16 | 2.08 |
| Mn-O$^{\delta 2}$(D159) | 2.04 | 2.05 | 2.15 | 2.11 | 2.03 | 2.00 |
| Mn-O$^{1}$(LIG) | 2.04 | 1.94 | 2.04 | 2.03 | 2.36 | 1.83 |
| **Distance** | | | | | | |
| Mn-O$^{2}$(LIG)[b] | 2.52 | 2.61 | 2.70 | 2.70 | 2.90 | 2.66 |

[a]Spin populations of DFT calculations are listed in Supplementary Table 4.
[b]The second oxygen of the dioxygen ligand is not directly bound to the Mn ion.
[c]Bond distances correspond to the chain *B* active site of D₂O₂-soaked Trp161Phe MnSOD.
[d]Geometry optimized with a Mn ion of S = 5/2, and a HO₂⁻ of S = 0.
[e]Geometry optimized with a Mn ion of S = 5/2, and a HO₂· of S = 1/2. Optimization with a HO₂· of S = −1/2 collapses to [Mn³⁺-⁻OOH].
[f]Geometry optimized with a Mn ion of S = 2 and a HO₂⁻ of S = 0.

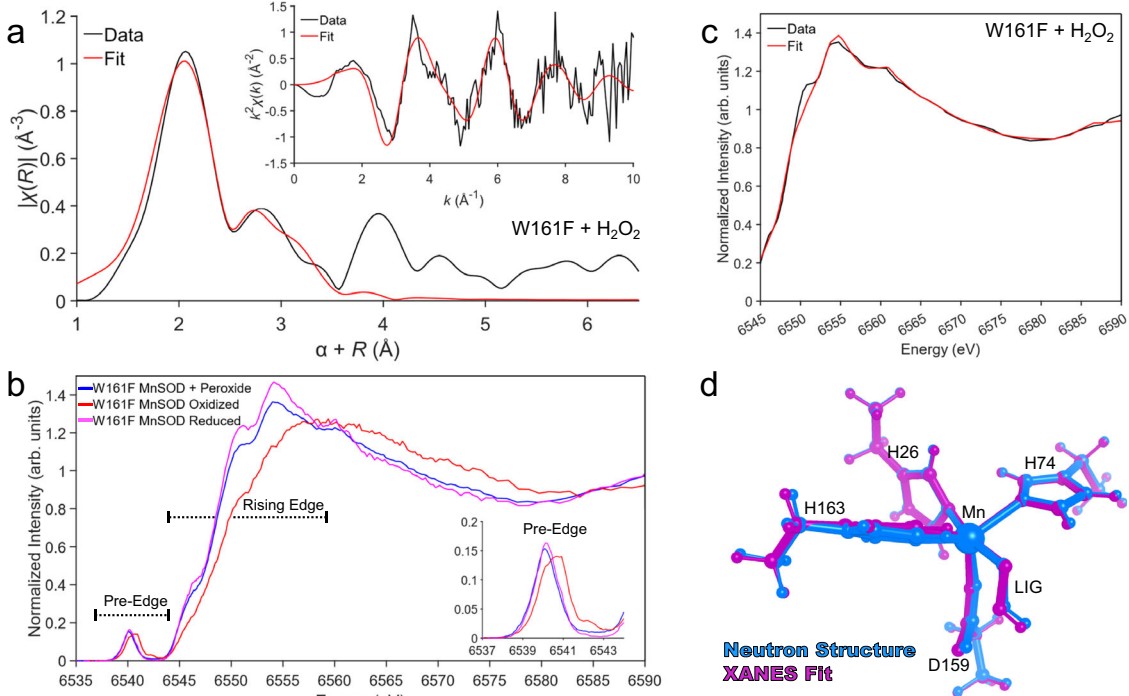

**Fig. 3 | X-ray Absorption Spectroscopy of Trp161Phe MnSOD. a** Fourier transform of Mn K-edge EXAFS data [$k^2\chi(k)$] from hydrogen peroxide-soaked Trp161Phe MnSOD with the raw EXAFS spectrum seen in the inset. Due to the scattering phase shift, the distance found by the Fourier Transformation (*R*) is -0.5 Å shorter than the actual distance, and a 0.5 Å correction (α) was implemented. The black line represents the experimental data, while the red line is simulated EXAFS spectra from the neutron structure fit to the experimental data. **b** Kα HERFD-XANES of Trp161Phe MnSOD treated with hydrogen peroxide, potassium dichromate, and

sodium dithionite to isolate the product-inhibited, Mn³⁺SOD resting, and Mn²⁺SOD resting states, respectively. The inset corresponds to a zoom-in of the pre-edge. **c** Fit of HERFD-XANES spectra of Trp161Phe MnSOD treated with hydrogen peroxide. The black line represents the experimental data, while the red line is simulated XANES spectra from the neutron structure fit to the experimental data. **d** Overlay of the neutron and XANES fit structures that are colored magenta and cyan, respectively. Data from this figure are provided within the Source Data file.

divalent Mn²⁺ ion (Table 2)[27]. The largest contributor to the second peak observed at -2.9 Å are seven carbon atoms that correspond to the C$^{\delta 2}$ and C$^{\epsilon 1}$ of the three Mn-bound histidines and the C$^{\gamma}$ of the Mn-bound aspartate (Supplementary Table 2). For the second oxygen of the dioxygen ligand (O², Table 2), the best distance fit for its scattering is at 2.52 Å, which is in slight contrast to the neutron structure and DFT distances of 2.62 Å and 2.70 Å, respectively. This difference may be due to the atom having more degrees of freedom than those directly bound to the Mn ion. Overall, the structural parameters of the Mn-bound atoms derived from the EXAFS spectrum agree with the crystallographic and QM counterparts.

We used Kα high-energy resolution fluorescence detected absorption (HERFD) to measure the XANES region. In K-edge XANES, the oxidation state and coordination geometry can be inferred from the spectra's rising edge and pre-edge regions (Fig. 3b). The energy of the rising edge increases as the oxidation state of the metal increases while staying in the same spin state. When comparing the rising edges in the XANES region between oxidized and reduced Trp161Phe MnSOD, a shift to lower energy is observed that indicates a one-electron difference of oxidation state and coincides with Mn³⁺SOD and Mn²⁺SOD resting forms, respectively (Fig. 3b)[68]. With hydrogen peroxide, the rising edge is most like the reduced counterpart, indicative

of a $Mn^{2+}$ ion. The feature around 6554 eV of peroxide-soaked Trp161Phe MnSOD is of lower intensity than the reduced form while a higher intensity is observed for peroxide-soaked Trp161Phe MnSOD at ~6570 eV. Additionally, a noticeable variation in shape resonance (6540–6590 eV) is seen between the two $Mn^{2+}$ complexes. These differences are caused by the replacement of the solvent molecule in reduced Trp161Phe MnSOD with a dioxygen species in peroxide-soaked Trp161Phe MnSOD. The HERFD-XANES spectra for wildtype MnSOD demonstrate the same trends; however, the dioxygen-bound species from peroxide soaking is not as well isolated (Supplementary Fig. 3). This is because wildtype has less product inhibition than Trp161Phe (Table 1). For the pre-edge region, the intensity (i.e., area of the pre-edge feature) corresponds to metal 1s to 3d transitions. These transitions gain intensity through 4p character mixing into the 3d orbitals from symmetry distortions and a loss of inversion symmetry[69–71]. A six-coordinate Mn complex is expected to have a different pre-edge intensity compared to a five-coordinate complex. For Trp161Phe with peroxide treatment, the pre-edge intensity is comparable to both the oxidized and reduced and indicates the coordination number is the same between the three chemical species (Fig. 3b). Like the rising edge, the pre-edge also shifts in energy depending on the oxidation state of the metal. A higher energy pre-edge for oxidized Trp161Phe MnSOD coincides with a $Mn^{3+}$ ion, while a lower energy pre-edge for reduced and peroxide-soaked Trp161Phe MnSOD represents a $Mn^{2+}$ ion. From the HERFD-XANES data, we can conclude that the introduction of hydrogen peroxide leads to a $Mn^{2+}$ ion that is five-coordinate, though it differs in structure compared to the reduced resting state.

Next, we performed structural refinement using the shape resonance of the HERFD-XANES spectra between 6545 and 6590 eV using combined QM chemistry simulations and machine learning[72,73]. With this method, systematic deformations of an input neutron structure were applied, and the corresponding XANES spectra of each deformation were simulated. The simulated spectra of the deformed structures then served as a training set for assigning a structure-to-spectra relationship for structural refinement. For the peroxide-soaked Trp161Phe complex, the simulated spectra of the best-fit structure capture nearly all features and relative intensities of the experimental spectra, with the exception of the 6550 eV feature that is found by the fit but underestimated (Fig. 3c, d). These trends are also the case for oxidized and reduced counterparts for the variant and wildtype (Supplementary Fig. 4). The Mn bond distances of the XANES structural refinement are in agreement with the structures from EXAFS, neutron diffraction, and DFT (Table 2 and Supplementary Table 3). Overall, the structural information of the inhibited complex extracted from the shape resonance of the HERFD-XANES spectra confirms the structure derived by the other methods.

Clear evidence for the oxidation state of the Mn during product inhibition has previously been challenging to obtain due to ambiguities in optical spectra, large zero-field splitting values for the Mn using conventional EPR, and efforts to capture the inhibited complex with the wildtype enzyme leading to a mixture of species[35,37,50–52,67]. Historically, the product-inhibited oxidation state has generally been accepted to be $Mn^{3+}$ as the oxidative addition of dioxygen species to low-valent metal ions has been described[26,35,37,50–52]. However, the HERFD data resemble a divalent Mn ion in both the rising edge and pre-edge regions with the introduction of hydrogen peroxide to the Trp161Phe variant (Fig. 3b). Furthermore, the metal bond distances observed by neutron diffraction and EXAFs support a divalent Mn ion (Table 2). Previous wildtype and Trp161Phe studies that coupled kinetics and visible spectra demonstrate that the inhibited complex lacks the characteristic 480 nm absorbance of the five-coordinate trivalent Mn oxidation state, although they were unable to rule out a six-coordinate $Mn^{3+}$ complex[26,50,51]. These past observations can be reconciled with our data that indicate a five-coordinate $Mn^{2+}$ complex.

It was surprising that $H_2O_2$ reduces the Mn ion. The reduction potential of wildtype $Mn^{3+}$SOD to $Mn^{2+}$SOD is 0.4 V, and the oxidation potential of $H_2O_2$ to $O_2^{\bullet-}$ is −0.85 V[66]. These values indicate that the reduction of $Mn^{3+}$SOD by $H_2O_2$ is not thermodynamically favorable. However, previous reports have noted that mixing $Mn^{3+}$SOD and stoichiometric excesses of $H_2O_2$ leads to a slow reaction yielding $Mn^{2+}$SOD[26,50–52]. The process was best fit by second-order kinetics, where reduction of $Mn^{3+}$SOD and $H_2O_2$ results in $Mn^{2+}$SOD and $O_2^{\bullet-}$ (a backward reaction), and then a subsequent reaction of $O_2^{\bullet-}$ with a second $Mn^{3+}$SOD to yield another $Mn^{2+}$SOD site (a forward reaction)[52]. While $H_2O_2$ reducing the Mn ion is thermodynamically unfavorable, past investigations have observed that excesses of $H_2O_2$ reduce the Mn ion.

Several studies have further observed that introducing high concentrations of $H_2O_2$ to $Mn^{3+}$SOD leads to the formation of the inhibited complex that decays to $Mn^{2+}$SOD[26,50–52]. For our neutron diffraction and XAS experiments, 0.28 M $D_2O_2/H_2O_2$ was used to ensure full occupancy and unambiguously isolate the product-inhibited Trp161Phe MnSOD complex in the crystal or in the 3 mM protein solutions used in XAS. Scanning stopped-flow spectrophotometry studies by Hearn et al. noted that mixing $H_2O_2$ and $Mn^{3+}$SOD at a 10:1 ratio led to rapid formation of the inhibited complex, though observation of a $O_2^{\bullet-}$ intermediate is lost during the dead time of the instrument (1.4 ms)[50]. Interestingly, these studies correlate product inhibition achieved through mixing $H_2O_2$ to the measured kinetics in the forward direction (i.e., reactions with $O_2^{\bullet-}$, Table 1). For example, variants with $k_2 \ll k_3$ (Table 1), like Trp161Phe, accumulate the inhibited complex with excessive molar ratios of $H_2O_2$. Indeed, mixing $Mn^{3+}$SOD with either $O_2^{\bullet-}$ or excessive concentrations of $H_2O_2$ leads to the same characteristic UV spectra of product inhibition[50]. These past observations suggest that the product-inhibited achieved through mixing $H_2O_2$ with $Mn^{3+}$SOD may be a combination of reverse and forward reactions.

Thus far, we have not addressed the electronic identity of the singly-protonated dioxygen ligand. Is it a protonated superoxide ($HO_2^{\bullet}$) or hydroperoxyl anion ($HO_2^-$) bound to $Mn^{2+}$ ion? At this time, we cannot experimentally differentiate between $HO_2^{\bullet}$ and $HO_2^-$. Given that $H_2O_2$ soaking leads to the reduction of the Mn ion[26,51,52], a $[Mn^{2+} \cdot {}^-OOH]$ complex could form. Our DFT calculations indicate that a high spin $[Mn^{2+} \cdot {}^-OOH]$ complex has a Mn-$O^1$(LIG) distance that is 0.32 Å or longer compared to distances found from our experimental methods (Table 2). Broken-symmetry DFT calculations of a low-spin $[Mn^{2+} \cdot {}^-OOH]$ complex collapses to a $[Mn^{3+} \cdot {}^-OOH]$ complex, which differs from our XAS data indicating a $Mn^{2+}$-containing complex (Fig. 3b). Another possibility is that a $H_2O_2$ molecule is deprotonated and binds the $Mn^{2+}$ ion. Our DFT calculations most closely resemble our experimental data when the identity of the dioxygen species is $HO_2^-$ (Table 2). $HO_2^-$ may be more chemically likely to form a stable product-inhibited complex due to its ionic rather than radical character. While we cannot definitively ascribe an electronic identity to the dioxygen species, our quantum chemistry calculations (Table 2) support the interpretation of an $HO_2^-$ molecule.

## Determination of the molecular orbitals that define redox reactivity

To investigate the molecular orbitals that determine the enzyme's activity, ground-state and excited-state DFT calculations were performed along with analysis of the HERFD-XANES pre-edge (Fig. 4). Three complexes were studied as they reflect the most physiologically relevant complexes that had experimental data: (1) wildtype $Mn^{3+}$SOD, (2) wildtype $Mn^{2+}$SOD, and (3) Trp161Phe $Mn^{2+}$SOD bound to $HO_2^-$. Unambiguous structural data of wildtype MnSOD bound to $HO_2^-$ is unavailable, so the Trp161Phe variant was used instead. For our calculations, we used high spin S = 2 and S = 5/2 for the $Mn^{3+}$ and $Mn^{2+}$ states, respectively, as indicated by experimental data[74].

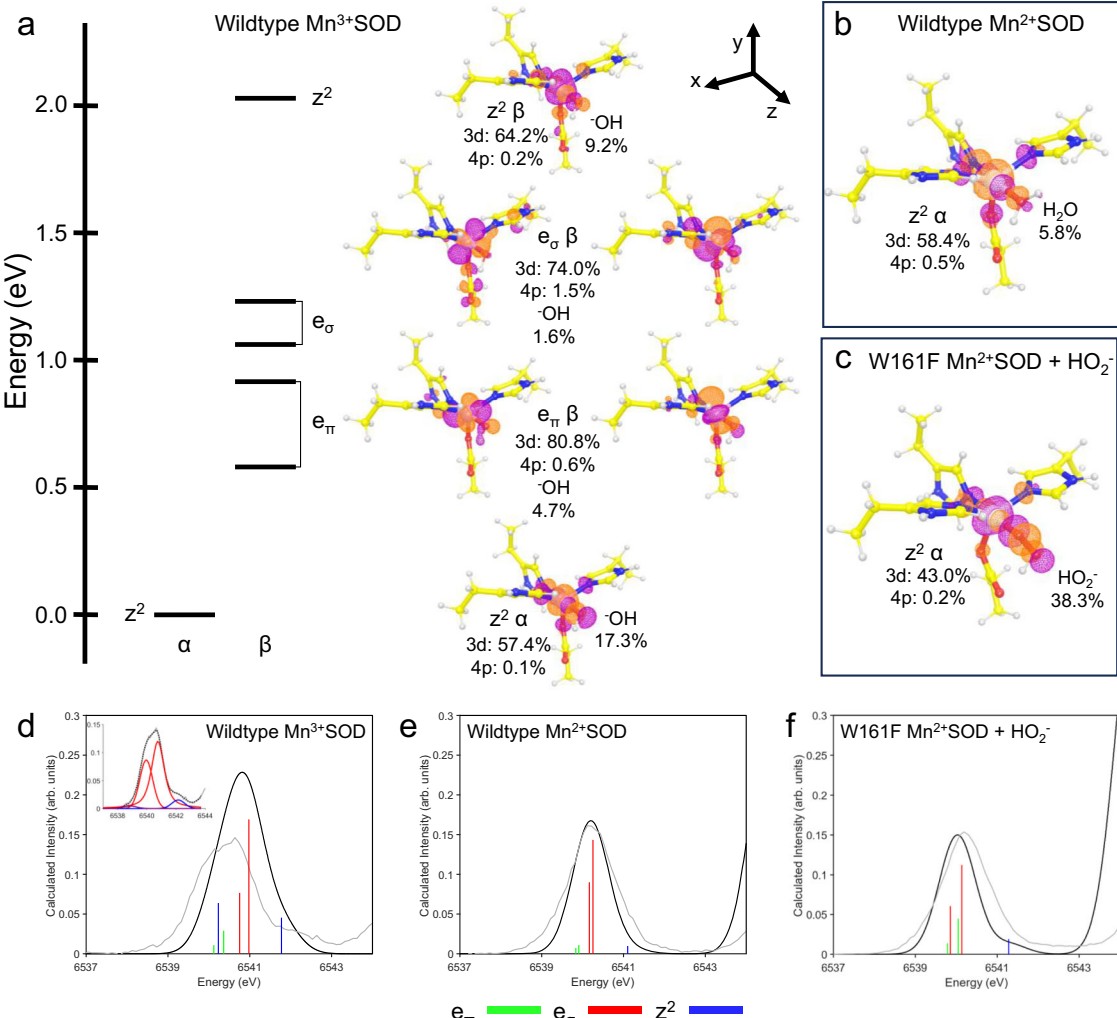

**Fig. 4 | The molecules orbitals involved in enzyme activity. a** Energy diagram of the unoccupied valence orbitals of Mn³⁺SOD. The energy axis is shifted to set the $z^2$ α orbital at zero. The isosurface plots of the unoccupied orbitals are contoured at 0.05 au. Percentages of orbital character are derived from Löwdin population analysis where the $C_{3v}$ symmetry-related xz and yz orbitals were averaged for the $e_\pi$ values, and the xy and x²-y² orbitals were averaged for the $e_\sigma$ values. **b** The HOMO of Mn²⁺SOD. **c** The HOMO of Trp161Phe Mn²⁺SOD bound to HO₂⁻. **d**–**f** TDDFT simulated spectra where the simulated spectrum is colored black, and the corresponding experimental HERFD-XANES spectrum is colored gray. The green, red, and blue vertical lines correspond to $e_\pi$, $e_\sigma$, and $z^2$ transitions, respectively. Simulated intensities were uniformly scaled to experimental intensities, and the energy axis was shifted. Due to a misestimation of the Mn³⁺SOD pre-edge spectra from the TDDFT simulation (see text), a pseudo-Voight peak fit was performed with identical color-coding of the transitions and is included as an inset for (**d**). Unclear peak identities are colored purple.

For the ground-state DFT calculations on Mn³⁺SOD, the unoccupied metal 3d orbitals largely determine the redox reactivity of the enzyme. The MnSOD active site has a distorted local $C_{3v}$ symmetry where the z-axis is along the Mn-O bond of the Mn-WAT1 interaction (Fig. 4a). High spin (S = 2) Mn³⁺ has one unoccupied α-spin orbital and five unoccupied β-spin orbitals. The lowest unoccupied molecular orbital (LUMO) is composed predominately of the $3d_{z^2}$ α orbital due to strong spin polarization from the partially occupied α-manifold and interacts strongly with ⁻OH. The LUMO is composed of 57% d character while the ⁻OH contribution to the orbital is 17%, indicating that the ⁻OH molecule is a contributor towards the reactivity of the orbital. The $e_\pi$ (xz/yz) orbitals of the β-manifold have strong d character of 81%, though are tilted along the axis suggesting mixing with other d orbitals. The similarly tilted $e_\sigma$ (xy/x²-y²) β orbitals have 74% d character and bond strongly with the amino acid ligands leading to 1.5% mixing of the Mn 4p orbitals with the $e_\sigma$ orbitals. Due to the 4p mixing, the $e_\sigma$ orbitals are expected to lead to significant electric dipole character even with these small percentages of mixing[69,70,75]. Lastly, the $3d_{z^2}$ β orbital is strongly σ-interacting with ligands along the z-axis. The 2.1 eV splitting

between the $z^2$ α and β orbitals is explained by spin polarization differences between the α- and β-manifolds reflected by a partially occupied α-manifold and an entirely unoccupied β-manifold. From the ground-state DFT calculations of Mn³⁺SOD, the $3d_{z^2}$ α orbital is expected to be the orbital participating in enzymatic redox reactions, and the ⁻OH molecule contributes towards reactivity.

Ground-state DFT calculations were performed to analyze the highest occupied molecular orbitals (HOMOs) of Mn²⁺SOD and Trp161Phe Mn²⁺SOD bound to HO₂⁻. The HOMOs are the orbitals participating in catalytic activity for Mn²⁺-containing complexes. In both complexes, the $3d_{z^2}$ α orbital is the HOMO and interacts with the H₂O or HO₂⁻ ligand along the z-axis (Fig. 4b, c). For wildtype Mn²⁺SOD, the d character of 58% strongly resembles that of the unoccupied $3d_{z^2}$ α orbital of Mn³⁺SOD with 57% d character (Fig. 4a). Given the protonation of the solvent ligand in Mn²⁺SOD (Fig. 4b), the 2p character from the solvent contributes less density to the $3d_{z^2}$ α orbital compared to that of Mn³⁺SOD. In the case of Trp161Phe Mn²⁺SOD bound by HO₂⁻, the HO₂⁻ contributes a large portion of density to the orbital, with Mn 3d and HO₂⁻ contributions of 43% and 38%, respectively (Fig. 4c). The

density localized on the $HO_2^-$ is reflective of the 2p π-antibonding orbital. Since the $z^2$ α orbital performs redox reactions, oxo or dioxo species involved in catalysis would bind in an orientation favoring interaction along the z-axis as it leads to the greatest orbital overlap. Analysis of the HOMOs potentially explains why $HO_2^-$ prefers to displace the metal-bound solvent rather than binding opposite the aspartate residue.

Next, we calculated the Mn K-pre-edge XAS for the three complexes using time-dependent density functional theory (TDDFT), an excited-state simulation, to assign the metal 3d orbital contributions to the experimental HERFD-XANES spectra of the pre-edge peak. The simulated spectra and corresponding experimental spectra are plotted in Fig. 4d–f. The vertical stick heights correspond to the intensity of the 1s to 3d transitions with the sum of quadrupole and dipole contributions. Overall, the spectral shapes of the divalent complexes (Fig. 4e, f) are well reproduced by the simulation. For the trivalent complex, the interpretation of the TDDFT simulation is trickier because the splitting of the orbital transitions is underestimated, while the intensity is overestimated (Fig. 4d).

The TDDFT simulation of wildtype $Mn^{3+}$SOD (Fig. 4d) identifies the $e_σ$ transitions as the primary contributors of intensity, though it has a misestimation of the overall spectral intensity and shape. Significant dipole intensity is found in the $e_σ$ and $z^2$ transitions. The $e_σ$ orbitals bear the most dipole character which is reflective of the higher 4p mixing seen in the ground-state DFT calculations (Fig. 4a). The valence 3d orbital splitting of the simulation is underestimated and the contribution of the transitions to the spectra is better represented by pseudo-Voigt peak fitting of the experimental data that supports an increased splitting energy (inset, Fig. 4d). Four peaks were required to fit the spectrum and attempts to fit five or more peaks did not lead to a better solution. The two larger peaks found in the fit potentially stem from the splitting of the $e_σ$ orbitals ($xy$/$x^2$-$y^2$) due to the distorted $C_{3v}$ symmetry where one of the orbitals may have more overlap with those of ligands. The higher energy peak found in the fit at ~6542.5 eV is best identified as the $z^2$ β transition supported by both ground and excited-state DFT. A fourth peak is capable of being fit as a small shoulder at ~6539 eV, though the orbital contribution to the peak is unclear as it could be of $e_π$ or $z^2$ α character (or both). The $e_π$ or $z^2$ α orbitals are probably underneath the stronger $e_σ$ orbitals. For the TDDFT simulation, the overestimation of the transition intensities and underestimation of the valence 3d orbital splitting has been previously noted for other high spin S = 2 $d^4$ complexes with similar symmetry[75,76]. In brief, the Mn 1s core hole exchange interaction with the valence 3d orbitals is overestimated by DFT, and its effects in the spectra are most prominent when there are both α and β transitions. This overestimation explains why the misestimations are less pronounced in the divalent S = 5/2 $d^5$ complexes (Fig. 4e, f) as they contain only β transitions. Nevertheless, the TDDFT simulation suggests that the $e_σ$ transitions are the predominant contributors to pre-edge spectral intensity for wildtype $Mn^{3+}$SOD.

For $Mn^{2+}$SOD, the assignment of the metal 3d orbital contributions to the experimental spectra is more straightforward. The TDDFT simulation of wildtype $Mn^{2+}$SOD (Fig. 4e) suggests the $e_σ$ orbitals have the most dipole character from 4p mixing and thus predominantly contribute to the intensity. The $e_π$ orbital transitions are mostly of quadrupole character, leading to a weak low-energy tail. Note that the $e_π$ orbitals are allowed to mix with the 4p orbitals, though have small effects due to π-bonding[71]. The $z^2$ transition consists of low dipole intensity leading to a high energy tail. The trend of transition energy ($e_π$, $e_σ$, $z^2$) mirrors that of $Mn^{3+}$SOD (Fig. 4d) and suggests the order of the metal 3d orbital energies is maintained between the two resting redox states. Overall, the TDDFT simulation of wildtype $Mn^{2+}$SOD matches the experimental data.

For the TDDFT simulation of Trp161Phe $Mn^{2+}$SOD bound to $HO_2^-$ (Fig. 4f), the overall spectra shape agrees well with that of the

experimental data. Here, all transitions bear significant dipole intensity, the $e_π$ and $e_σ$ transitions overlap in energy, and the $z^2$ transition contributes a tail to the spectra. The observations coincide with further distortion of $C_{3v}$ symmetry introduced from the binding of the $HO_2^-$ molecule, causing the metal 3d orbitals to mix further. This mixing distributes the dipole character over the five unoccupied β orbitals, leading to an overlap of the $e_π$ and $e_σ$ sets of orbitals. Importantly, the mixing permits the distinction of a dioxygen species bound complex over that of the solvent molecule-bound complexes seen in $Mn^{3+}$ and $Mn^{2+}$ resting states (Fig. 4d, e). Overall, the TDDFT simulation of Trp161Phe $Mn^{2+}$SOD bound to $HO_2^-$ agrees with the experimental data.

The ground-state and excited-state DFT simulations help explain the catalytic mechanism of MnSOD. The S = 2 $d^4$ configuration of $Mn^{3+}$SOD and the distorted trigonal ligand field leads to spin polarization of the $3d_{z^2}$ α orbital and its large splitting with the $3d_{z^2}$ β orbital (Fig. 4a). This splitting is reflected in the experimental pre-edge spectra though is underestimated in the TDDFT simulations and is documented by studies of similar complexes (Fig. 4d)[75,76]. Importantly, the splitting leads to the $3d_{z^2}$ α becoming the LUMO that is the putative electron acceptor orbital for reaction with $O_2^{\bullet-}$. The $3d_{z^2}$ α orbital mixes strongly with antibonding molecular orbitals of the metal-bound ligands. In particular, the $^-$OH molecule bound along the z-axis contributes a relatively large portion of density towards the LUMO, suggesting that it is a determinant in redox activity. For $Mn^{2+}$SOD that has an S = 5/2 $d^5$ configuration, ground-state DFT calculations (Fig. 4b) indicate that the $3d_{z^2}$ α orbital is the HOMO, the putative electron donor orbital for reaction with $O_2^{\bullet-}$. The TDDFT simulation (Fig. 4e) matches the HERFD-XANES spectra to suggest that our calculations are recapturing the experimental data for the resting $Mn^{2+}$ complex. For the Trp161Phe $Mn^{2+}$SOD complex bound to $HO_2^-$, the HOMO is again the $3d_{z^2}$ α orbital though the 2p π-antibonding character of $HO_2^-$ contributes a large portion of density to the orbital (Fig. 4c). Jackson and colleagues have previously hypothesized that dioxygen species may preferentially displace a metal-bound solvent for Mn ion binding since the dioxygen π-antibonding orbitals would overlap with the Mn $3d_{z^2}$ α orbital[77]. Additionally, the experimental and TDDFT simulations of the pre-edge (Fig. 4f) permit spectral discrimination of a dioxygen-bound $Mn^{2+}$SOD complex against a resting state $Mn^{2+}$SOD complex. Altogether, the DFT simulations indicate that (1) the $z^2$ α orbital is the orbital performing redox reactions, (2) reactivity is influenced by the species bound along the z-axis of the Mn, and (3) displacement of the solvent molecule for dioxygen species binding is chemically preferred due to orbital overlaps.

## Unexpected second-sphere hydrogen peroxide binding site

We previously demonstrated that residues Gln143, Tyr166, His30, and Tyr34 have unusual $pK_a$s and undergo changes in protonation states during the MnSOD PCET cycle (Fig. 1)[27]. The active site gains two protons during the $Mn^{3+} \rightarrow Mn^{2+}$ redox transition where one proton is acquired by Tyr34 and the other by His30. These protons are lost during the $Mn^{2+} \rightarrow Mn^{3+}$ transition, which led us, at the time, to hypothesize that Tyr34 and His30 bind and protonate $O_2^{2-}$ to $H_2O_2$. This was supported by the fact that the gateway between these two residues is the only solvent-accessible entrance/exit of the active site. When investigating the active site of the $D_2O_2$-soaked structure at chain $B$, a strong omit $|F_o| - |F_c|$ difference density peak is seen between His30 and Tyr34 with an oblong shape that is longer than a water molecule (orange density, Fig. 5a). This is the same active site where the dioxygen ligand is bound to the $Mn^{2+}$ ion (Fig. 2a). We interpret the omit nuclear density as $D_2O_2$ (denoted as PEO, Fig. 5a) given (1) the crystal was soaked with $D_2O_2$ before data collection, (2) modeling and refining a $D_2O$ (water) molecule leads to unoccupied residual density (Supplementary Fig. 5), and (3) the density has two protruding features indicating two protons. The locations of the $D_2O_2$ protons are further

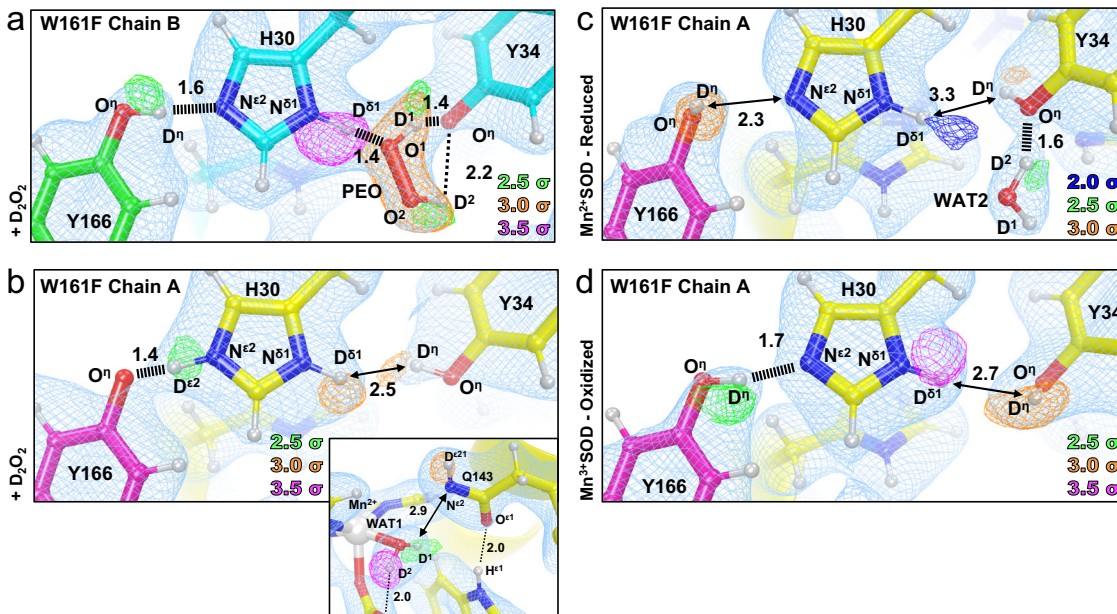

**Fig. 5 | Neutron structures and protonation states of second sphere active site residues in D$_2$O$_2$-soaked, reduced, and oxidized Trp161Phe MnSOD. a** D$_2$O$_2$-soaked Trp161Phe MnSOD at the active site of chain *B*. **b** D$_2$O$_2$-soaked Trp161Phe MnSOD at the active site of chain *A*. Inset highlights the structure near the divalent Mn ion. Chain *B* is more accessible to solvent than chain *A* and helps explain differences in ligand binding. **c** Divalent resting state of Trp161Phe MnSOD at the active site of chain *A*. **d** Trivalent resting state of Trp161Phe MnSOD at the active site of chain *A*. Blue, green, orange, and magenta omit |$F_o$| - |$F_c$| difference neutron scattering length density of protons are displayed at 2.0 σ, 2.5σ, 3.0σ, and 3.5σ, respectively. Light blue 2|$F_o$| - |$F_c$| density is displayed at 1.0σ. Distances are in Å. Dashed lines indicate typical hydrogen bonds and hashed lines indicate SSHBs, hydrogen bonds <1.8 Å. For the resting state structures, only one active site is shown due to high structural similarities, see Supplementary Fig. 2 for the other active site. All neutron structures were solved to 2.3 Å resolution.

verified by their nuclear density peaks (green density of PEO, Fig. 5a), with the D$^1$(PEO) proton forming a 1.4 Å SSHB with a deprotonated Tyr34. The D$^2$(PEO) forms a longer 2.2 Å hydrogen bond with anionic Tyr34 and may explain the unusual 17° dihedral angle of PEO. Furthermore, the D$_2$O$_2$ molecule forms a second 1.4 Å SSHB between atoms D$^{δ1}$(His30) and O$^1$(PEO). The proton between O$^η$(Tyr166) and N$^{ε2}$(His30) that we previously observed participating in a LBHB for the resting Mn$^{2+}$SOD structure (Fig. 1g) is instead fully localized onto O$^η$(Tyr166) and participates in a 1.6 Å SSHB with N$^{ε2}$(His30). The experimental data suggest a strong network of hydrogen bonds is present at physiological pH involving Tyr34-PEO-His30-Tyr166 and support our hypothesis that Tyr34 and His30 bind and protonate O$_2^{2-}$ to H$_2$O$_2$.

The active site of chain *A* for the D$_2$O$_2$-soaked Trp161Phe MnSOD, which does not have binding of a dioxygen species in either the first or second spheres, was next investigated (Fig. 5b). The active site has a divalent Mn$^{2+}$ ion with Mn bond distances and protonation states reflecting that of Trp161Phe and wildtype Mn$^{2+}$SOD (Supplementary Table 1). Without treatment of redox reagents or D$_2$O$_2$/H$_2$O$_2$, human MnSOD is ~90% trivalent[66] (Fig. 2e and Supplementary Fig. 3) and suggests peroxide reduced the metal ion, and the resulting species vacated the active site before flash-freezing. Here, it is also evident that the variant destabilized the Gln143-WAT1 interaction as they are not in a hydrogen bond (Fig. 5b, inset). The omit |$F_o$| - |$F_c$| difference density shows that Tyr34 is protonated (Fig. 5b) and contrasts with the deprotonated Tyr34 of chain B, supporting the idea that Tyr34 may gain a proton from D$_2$O$_2$. For His30 of chain A, omit |$F_o$| - |$F_c$| difference density is observed for protons on both nitrogen atoms at contours of 2.5σ or higher, indicating His30 is positively charged and Tyr166 is negatively charged. The D$^{ε2}$(His30) proton forms a 1.4 Å SSHB with O$^η$(Tyr166) and also differs with chain B, where the proton is localized on His30 rather than Tyr166. Given that the proton between N$^{ε2}$(His30) and O$^η$(Tyr34) is seen altering its position, a back-and-forth proton

transfer may occur between the heteroatoms depending on the ligand environment, such as the binding of a dioxygen species. The observation also reinforces the model originated from the room temperature wildtype neutron structures where a proton is either transiently shared between His30 and Tyr166 or a back-and-forth proton shuffle occurs during the catalytic cycle[27]. Overall, the neutron structure of D$_2$O$_2$-soaked Trp161Phe MnSOD at the active site of chain *A*, in conjunction with the active site of chain *B*, provides insight into the role of Tyr34, His30, and Tyr166 during product binding.

We next sought mechanistic clues from the neutron structures of resting state Trp161Phe Mn$^{2+}$SOD and Trp161Phe Mn$^{3+}$SOD. Both chains of each structure have high structural similarity (Supplementary Fig. 2). For Trp161Phe Mn$^{2+}$SOD, the protonation states of Tyr34, His30, and Tyr166 resemble those of the wildtype Mn$^{2+}$SOD though there are notable structural features. Tyr34 engages in a 1.6 Å SSHB with WAT2 where O$^η$(Tyr34) acts as a hydrogen bond acceptor (Fig. 5c). It is apparent that Tyr34 forms SSHBs with water molecules regardless of its protonation state and Mn oxidation state, supporting the notion that Tyr34 is central to proton turnover during catalysis[27]. Indeed, its mutation predominately affects the Mn$^{2+}$ to Mn$^{3+}$ redox transition of the enzyme[35,37,48]. Besides Tyr34, the other notable structural feature is the hydroxyl proton of Tyr166 that points away from His30 (Fig. 5c). It is unclear whether the orientation of the proton has catalytic significance, is somehow related to the Trp161Phe variant, or is a consequence of cryocooling. Regardless, the Trp161Phe Mn$^{2+}$SOD and the D$_2$O$_2$-bound neutron structures (Fig. 5a) provide evidence that Tyr34 is involved in SSHBs with both D$_2$O$_2$ and water molecules and supports the interpretation that Tyr34 is a central player in proton transfer.

Surprisingly, Tyr34 and His30 are both protonated in the neutron structure of Trp161Phe Mn$^{3+}$SOD, which is the opposite of wildtype Mn$^{3+}$SOD protonation states (Fig. 1d, f). Strong omit |$F_o$| - |$F_c$| difference density is observed for D$^η$(Tyr34), D$^{δ1}$(His30), and D$^η$(Tyr166) (Fig. 5d). A potential explanation for the different protonation states

 

are alterations of residue $pK_a$s due to small changes in the active site. Past studies across different MnSOD isoforms have established that subtle changes in hydrogen bonding and residue orientation lead to significant changes in catalysis, mainly due to shifts of the net electrostatic vectors[35,37,64,78,79]. Other changes in the Trp161Phe variant include the previously discussed Gln143 and WAT1 molecules (Fig. 2f) and a slight 1.7 Å movement of Tyr34 toward His30 (Supplementary Fig. 6). Our neutron diffraction data of Trp161Phe $Mn^{3+}$SOD suggest that subtle movement of active site residues affect the $pK_a$ of residues involved in proton transfer.

Our Trp161Phe MnSOD neutron structures indicate that $D_2O_2$ forms strong hydrogen bonds with His30 and Tyr34 and that residues Tyr34, His30, and Tyr166 undergo protonation changes. While the $D_2O_2$ binding site between His30 and Tyr34 was predicted, the neutron structure is the first experimental evidence for this binding[27,34,35,37,64,80]. The binding site has been thought to be where an $O_2^{2-}$ or $HO_2^-$ species gain protons during the $Mn^{2+}$ to $Mn^{3+}$ redox transition ($k_2$). However, our neutron structure of $D_2O_2$-soaked Trp161Phe MnSOD brings the possibility that it may also play a role during product inhibition, with three possible interpretations of the data. First, $D_2O_2$ binding occludes the only entrance to the active site and may potentially block substrate from entering since the Tyr166-His30-PEO-Tyr34 network is composed of several SSHBs (Fig. 5a). Second, the life of the network is short-lived, and a PEO → Tyr34 proton transfer would create a $HO_2^-$ with high electrostatic affinity towards the Mn ion. Third, a $HO_2^-$ molecule already bound to $Mn^{2+}$ may stabilize $H_2O_2$ species binding between His30 and Tyr34 and block the product from exiting. Altogether, the neutron structures provide considerable insight into MnSOD catalysis by revealing both binding orientations of dioxygen species and protonation states.

## Summary of active site configurations solved

The neutron structures, XAS data, and simulations of $D_2O_2$-soaked reduced, and oxidized Trp161Phe MnSOD present four primary configurations of active sites. For chain $B$ of the $D_2O_2$-soaked neutron structure, two dioxygen species are found within the active site (Fig. 6a). One of the dioxygen species is singly-protonated and bound directly to the metal ion, replacing the position of WAT1, and forming a SSHB with Gln143 that is in the neutral form (Fig. 2a). Our XAS measurements and DFT simulations support the interpretation that the Mn ion is divalent and the singly-protonated dioxygen species is $HO_2^-$ (Figs. 3 and 4, Table 2). In the same active site, $D_2O_2$ is observed binding tightly between an ionic Tyr34 and neutral His30 (Fig. 5a). For chain $A$ of the $D_2O_2$-soaked neutron structure, the active site is reduced, but a dioxygen ligand is not present, and Gln143 is deprotonated (Fig. 6b). Since high peroxide concentrations reduce the Mn ion (Fig. 3b)[26,50,51], we presume that the active site metal ion underwent reduction and resulting dioxygen species left before cryocooling (Fig. 5b). For the Trp161Phe reduced resting state, both chains have the same structure, and WAT1 is forming a hydrogen bond with the amide anion form of Gln143 (Fig. 6c). For the Trp161Phe oxidized resting state, the structure is the also same between chains where WAT1 is $^-$OH and Gln143 is in the canonical amide form (Fig. 6d). Overall, our data indicate that a $H_2O_2$ molecule undergoes deprotonation prior to binding $Mn^{2+}$, and His30, Tyr166, Gln143, and Tyr34 are capable of changing protonation states.

Altogether, the present work reveals key features of product inhibition and the related active site proton environment. Through combined neutron diffraction, XAS, and computational chemistry calculations, we find that (1) the inhibited complex is a five-coordinate $Mn^{2+}$ complex where a $HO_2^-$ replaces WAT1 found in the resting states; (2) how easily WAT1 may be displaced correlates with the extent of product inhibition; (3) preference for dioxygen species binding in place of the WAT1 position is a result of the overlap with the Mn $3d_{z}^2$ orbital; (4) $H_2O_2$ strongly binds to an anionic Tyr34 and His30; (5) an

anionic Tyr166 and cationic His30 forms at least transiently with $H_2O_2$ treatment, and (6) slightly different orientations of active site residues can alter their $pK_a$s due to the strong electrostatic vectors provided by the Mn ion. With this knowledge of product binding, we propose a mechanism for how product inhibition and relief may occur.

## Mechanism for MnSOD product inhibition and relief

From our use of high $H_2O_2/D_2O_2$ concentrations, we postulate that the inhibited complex we observe is from first a backward reaction of $H_2O_2$ with $Mn^{3+}$ to form a $O_2^{\bullet-}$ species and $Mn^{2+}$, and then a forward reaction of $O_2^{\bullet-}$ with another $Mn^{3+}$ site in the presence of $H_2O_2$ to form the inhibited complex. Here, $H_2O_2$ is seen as a means to generate $O_2^{\bullet-}$ and study the forward reactions. We have framed our proposed mechanism only in the forward direction to better represent what would occur in physiological conditions.

We propose a mechanism of product inhibition dependent on a molecule of $O_2^{\bullet-}$ arriving and reducing $Mn^{3+}$ before the departure of a recently formed $H_2O_2$. $O_2^{\bullet-}$ reduces $Mn^{3+}$ to $Mn^{2+}$ in conjunction with anionic Tyr34 accepting a proton from $H_2O_2$ to form neutral Tyr34 and $HO_2^-$ (Fig. 7a). Note that reaction of $Mn^{3+}$ with $O_2^{\bullet-}$ to form $Mn^{2+}$ and $O_2$ is represented by a gain of an electron due to the lack of experimental evidence for whether $O_2^{\bullet-}$ requires coordination to the Mn ion for electron transfer. After the PCET, $HO_2^-$ immediately displaces WAT1 to bind the $Mn^{2+}$ ion and forms a SSHB with Gln143 to make the inhibited complex (Fig. 7b). Decay of the inhibited complex involves a Gln143 to $HO_2^-$ proton transfer that yields a Gln143 in the amide anion form, and the $H_2O_2$ product is subsequently replaced by a water molecule (Fig. 7c, d).

Altogether, the proposed mechanism of product inhibition is dependent on the presence of $H_2O_2$ within the active site during the $Mn^{3+}$ to $Mn^{2+}$ redox transition, deprotonation of $H_2O_2$ by Tyr34, and subsequent displacement of WAT1 by $HO_2^-$. Inhibition is characterized by the elimination of the back-and-forth WAT1 and Gln143 proton transfer that is central for PCET catalysis[27,67], where dioxygen species replacing the WAT1 position is a result of favored molecular orbital overlap with the Mn $3d_{z}^2$ α orbital. Relief of the complex involves the protonation of $HO_2^-$ to form $H_2O_2$, which is replaced by a water molecule. The combination of neutron diffraction and XAS has revealed, to our knowledge, the first direct experimental evidence for the identity of the inhibited complex and how it is formed and relieved. Most published mechanistic models of MnSOD product inhibition have presumed that inhibition proceeds through a $Mn^{2+}$SOD and $O_2^{\bullet-}$ reaction ($k_3$, Table 1)[18,35,37], without the involvement of the Gln143 proton transfer, and independent of the $Mn^{3+}$ → $Mn^{2+}$ half-reaction, though our data are best described by the contrary. We also show with cryo-neutron crystallography that Tyr166 and His30 are unambiguously capable of being ionized (Fig. 5b), which presents further evidence that the active site metal ion environment leads to unusual $pK_a$s and biochemical states. In total, we conclude that human MnSOD achieves product inhibition through an unusual PCET mechanism. High product inhibition is achieved from the precise arrangement of active site residues and likely arose evolutionarily to regulate the many roles $H_2O_2$ plays in human cell physiology[35,37,47]. Abnormalities in cellular $H_2O_2$ concentrations are hallmarks of disease[43,44], and MnSOD product inhibition helps prevent cellular dysfunction.

## Methods

### Perdeuterated expression and purification

For deuterated protein expression of MnSOD, the pCOLADuet-1 expression vector harboring full-length cDNA of *MnSOD* was transformed into *E. coli* BL21(DE3) cells. Transformed cells were grown in $D_2O$ minimal media within a bioreactor vessel using $D_8$-glycerol as the carbon source[81]. Induction was performed with 1 mM isopropyl �-D-thiogalactopyranoside, 8 mM $MnCl_2$, and fed $D_8$-glycerol until an $OD_{600}$ of 15.0. Expression was performed at 37 °C for optimal Mn ion

 

**Fig. 6 | Summary of active site configurations observed for $D_2O_2$-soaked, reduced, and oxidized Trp161Phe MnSOD. a** Dioxygen-bound Trp161Phe MnSOD is seen in chain B. The singly-protonated dioxygen species bound to the metal ion (Fig. 2b) is probably hydroperoxyl anion as supported by our XAS measurements (Fig. 3) and DFT calculations (Fig. 4, Table 2). A $D_2O_2$ molecule is also coordinated between His30 and Tyr34 with SSHBs (Fig. 5a). **b** Active site of Trp161Phe MnSOD chain A was treated with $D_2O_2$, though cryocooling did not capture any dioxygen species (Fig. 5b). **c** Reduced resting state of Trp161Phe MnSOD (Figs. 2c, 5c). **d** Oxidized resting state of Trp161Phe MnSOD (Figs. 2e, 5d). All neutron structure active sites are shown in Supplementary Fig. 2. Dashed lines represent normal hydrogen bonds, and wide dashed lines are SSHBs. The portrayal of structures and bond lengths in 2D are not representative of those seen experimentally in 3D.

incorporation[82]. Harvested cell pastes were stored at −80 °C until purification. For protein purification (with hydrogenated reagents), cells were resuspended in a solution of 5 mM $MnCl_2$ and 5 mM 3-(N-morpholino)propanesulfonic acid (MOPS), pH 7.8. Clarified lysate was incubated at 55 °C to precipitate contaminant proteins that were subsequently removed by centrifugation. Next, soluble protein was diluted with an equal volume of 50 mM 2-(N-morpholino)ethanesulfonic acid (MES) pH 5.5, yielding a final concentration of 25 mM. Measurement of pH verified a value of 5.5 after dilution. Protein was applied onto a carboxymethyl sepharose fast flow column (GE Healthcare) and eluted with a sodium chloride gradient that contained 50 mM MES pH 6.5.

**Crystallization**

Perdeuterated MnSOD crystals were grown in either a microgravity environment aboard the International Space Station (ISS) or in an earthly environment at the UNMC Structural Biology Core Facility with hydrogenated reagents. For microgravity crystal growth, crystals were grown in Granada Crystallization Boxes (GCBs, Triana) through capillary counterdiffusion using fused quartz capillary tubes (VitroCom) that had inner diameters of 2.0 mm and outer diameters of 2.4 mm[83]. Crystals were grown from a 25 mg ml⁻¹ protein-filled capillary that was plugged with 40 mm of 2% agarose (w/w) and inserted into a GCB subsequently filled with precipitating agent consisting of 4 M potassium phosphate, pH 7.8. The pH of the phosphate buffer was achieved through 91:9 ratios of $K_2HPO_4$:$KH_2PO_4$. The GCBs were delivered to the ISS by SpX-17 as part of the *Perfect Crystals* NASA payload and returned to Earth 1 month later on SpX-18. The crystals within GCBs were observed to be resilient against travel damage and were placed within carry-on baggage during further aircraft travels to the UNMC Structural Biology Core Facility and ORNL. Further details of microgravity crystallization were described previously[84].

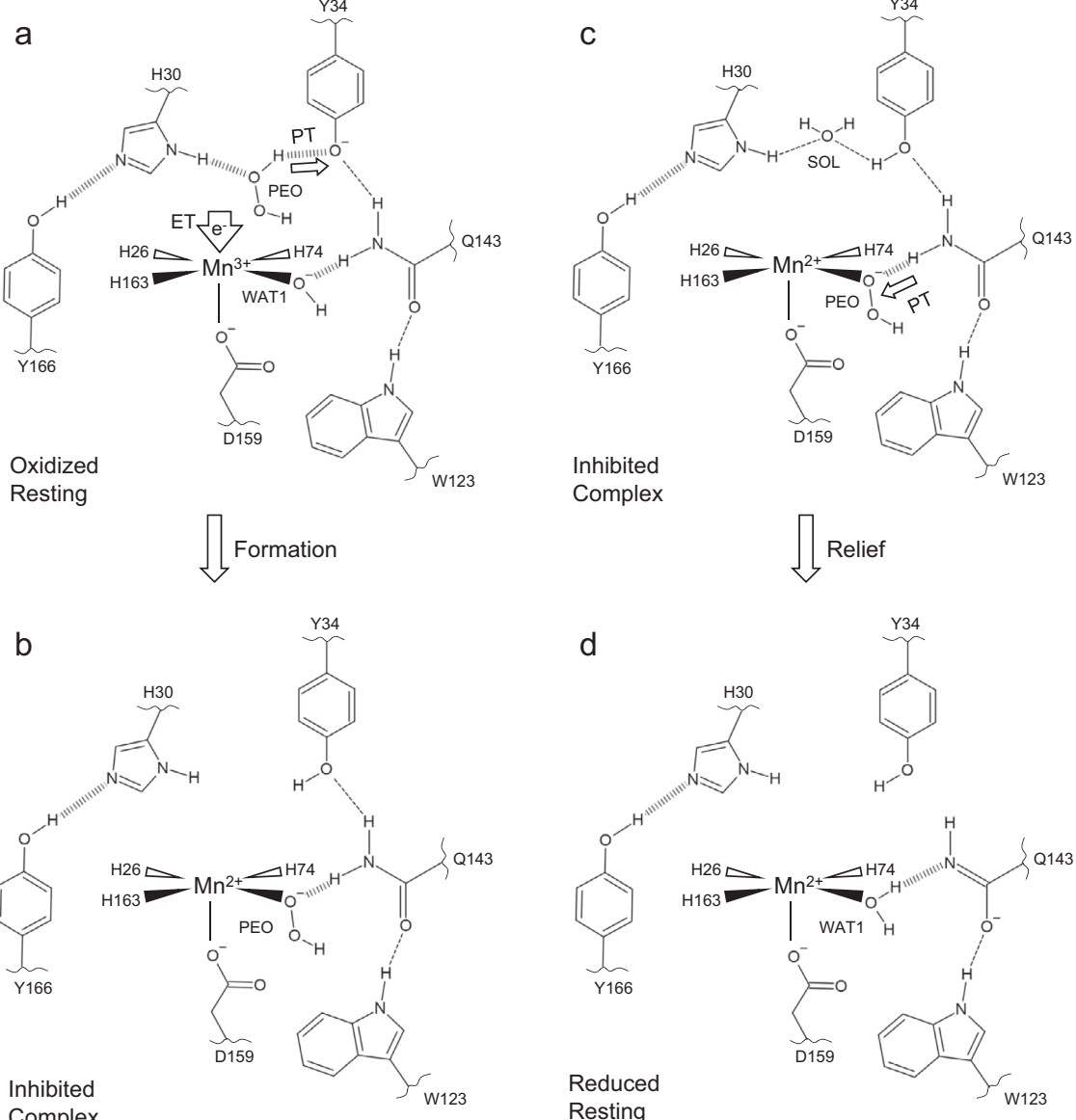

**Fig. 7 | A suggested mechanism for MnSOD product inhibition and relief.**
**a** Product inhibition is dependent on the presence of $H_2O_2$ (denoted as PEO) coordinated between His30 and Tyr34 during the $Mn^{3+} \rightarrow Mn^{2+}$ redox transition. Due to the lack of experimental evidence for $O_2^{\bullet-}$ binding and uncertainty in whether it requires coordination with the Mn ion for redox catalysis, the redox reaction is instead represented by a gain of an electron. For the formation of the inhibited complex to proceed, the gain of an electron by $Mn^{3+}$ coincides with the deprotonation of $H_2O_2$ by Tyr34. Note that His30 has been shown to change protonation states on both of its nitrogen atoms and could potentially extract a proton from $H_2O_2$ instead of Tyr34. **b** After the PCET, $HO_2^-$ replaces the WAT1 solvent molecule

to form the inhibited complex characterized by the elimination of a Gln143-WAT1 interaction while the Mn ion is in the divalent redox state. **c** The relief of the inhibited complex involves protonation of $HO_2^-$ by Gln143 to form $H_2O_2$ and an ionized Gln143 and subsequent replacement of the original WAT1 position by a water molecule. **d** After $H_2O_2$ leaves the active site, the $Mn^{2+}$SOD is formed that is characterized by an ionized Gln143 forming a SSHB with WAT1, and Tyr34, His30, and Tyr166 in the neutral states. Dashed lines represent normal hydrogen bonds, and wide dashed lines are SSHBs. The portrayal of the displayed structures and bond lengths in 2D are not representative of those seen experimentally in 3D.

For earthly crystal growth, crystallization was performed using a 9-well glass plate and sandwich box setup (Hampton Research), and the reservoir solution consisted of 1.9 M potassium phosphate adjusted to pH 7.8 by varying ratios of $KH_2PO_4$ and $K_2HPO_4$. The crystallization drop was a mixture of 60 μL of 23 mg mL$^{-1}$ concentrated protein solution (within a buffer of 50 mM MES pH 6.5) and 40 μL of the reservoir solution. Crystals grew up to 0.5 mm$^3$ after 6 weeks at 23 °C.

**Crystal manipulations**
Initial deuterium exchange was performed one of two ways, depending on the crystal growth conditions. For microgravity-grown crystals,

samples were placed in 1 mL of hydrogenated 4 M potassium phosphate pH 7.8. Deuterium was introduced with 0.1 mL incremental additions every 2 min of 4 M deuterated potassium phosphate ($K_2DPO_4$:$KD_2PO_4$) pD 7.8 (calculated by adding 0.4 to the measured pH reading) for a total of five times and a net volume addition of 0.5 mL. After 10 min, 0.5 mL of the solution was removed leading to a 1 mL solution consisting of 33% deuterium. The process is repeated enough times to gradually increase the deuterium content to -100%. The 4 M deuterated potassium phosphate also served as the cryoprotectant for the cryocooling process. Further details of the process were published[59]. For the crystals grown on earth, the initial deuterium exchange of crystals was performed by vapor diffusion in quartz

capillaries using deuterated solutions of 2.3 M deuterated potassium phosphate pD 7.8. For cryoprotection, the concentration of the deuterated potassium phosphate was incrementally increased within the capillaries until a concentration of 4 M was achieved.

For redox manipulation, the deuterated potassium phosphate solutions were supplemented with either 6.4 mM potassium permanganate ($KMnO_4$) to achieve the $Mn^{3+}$ oxidation state or 300 mM sodium dithionite ($Na_2S_2O_4$) to achieve the $Mn^{2+}$ state. Crystals were either sealed in capillaries or in 9-well glass plates to ensure maintenance of the desired oxidation state. For the Trp161Phe structure soaked with $D_2O_2$, redox reagents were not used. The dioxygen-bound complex was achieved by supplementing the cryoprotectant that the crystal was immersed in with $D_2O_2$ at a final concentration of 1% w/$v$ (0.28 M) and soaking for 5 min before cryocooling. Flash-cooling was performed with an Oxford diffraction cryostream[85]. Further details of ligand cryotrapping were published[59].

## Crystallographic data collection

Time-of-flight, wavelength-resolved neutron Laue diffraction data were collected from perdeuterated crystals using the MaNDi instrument[86,87] at the Oak Ridge National Laboratory Spallation Neutron Source with wavelengths between 2 and 4 Å. Sample sizes ranged from 0.3 to 0.6 $mm^3$ and data were collected to 2.30 Å resolution or better (Supplementary Table 5). Crystals were held in stationary positions during diffraction and successive diffraction frames were collected along rotations of the Φ axis. X-ray diffraction data were collected from crystals grown in conditions identical to those used for neutron diffraction using our Rigaku FR-E SuperBright home source (Supplementary Table 5). The $D_2O_2$-soaked and reduced Trp161Phe MnSOD neutron datasets were collected from crystals grown in microgravity, while all other data sets were from crystals grown on Earth.

## Crystallographic data processing and refinement

Neutron data were integrated using the MANTID software package v6.0.0[88–90] and wavelength-normalized and scaled with LAUENORM from the Daresbury Laue Software Suite v6.0[91]. X-ray diffraction data were processed using HKL-3000 v717[92]. Model building was performed in COOT v0.9.6[93,94]. Refinements of both neutron and X-ray models were completed separately with PHENIX.REFINE from the PHENIX suite v1.21[95]. The refinements were intentionally performed separately due to the known perturbations that X-rays have on the solvent structure, metal redox state, and metal coordination[28,96]. The X-ray model was first refined against its corresponding data set and subsequently used as the starting model for neutron refinement. Torsional backbone angle restraints were derived from the X-ray model and applied to neutron refinement using a geometric target function with PHENIX.REFINE[95]. Mn-ligand restraints for neutron refinement were derived from DFT calculations rather than the X-ray model to remove any influence of photoreduction. The neutron refinement was performed by modeling the D atoms of the active site last to limit phase bias. Initial rounds of refinement to fit protein structure included only non-exchangeable D atoms, which have stereochemical predictable positions. Afterward, H/D atoms were modeled onto the position of each amide proton, and occupancy was refined. In general, the asymmetric units of the neutron crystal structures had a deuterium content of ~85% for the amide backbone, and areas with low deuterium exchange (<50%) coincided with the presence of hydrogen bonds forming a secondary structure. Next, exchangeable proton positions of residues outside the active site (e.g., the hydroxyl group of serine/tyrosine) were manually inspected for obvious positive omit $|F_o| - |F_c|$ neutron scattering length density at a contour of 2.5σ or greater and modeled as a fully occupied deuterium. If the density was not obvious, and there was no chemically sensible reason for the residue to be deprotonated (which is the

case for residues outside the active site), the proton position was H/D occupancy refined. $D_2O$ molecules outside the active site were then modeled and adjusted according to the nuclear density. Last, D atoms of the active site were modeled manually. At the active site, a residue is considered deprotonated when (1) attempts to model and refine a proton result in negative $|F_o| - |F_c|$ difference neutron scattering length density, (2) all the other protons of the residue can be placed, and (3) the heavy atom that is deprotonated acts as a hydrogen bond acceptor. As chemically ideal covalent bond distances of D atoms were ensured during model building and refinement, small deviations from D atom positions and omit $|F_o| - |F_c|$ neutron scattering length density centers were expected from the data resolution (2.3 Å).

## X-ray absorption spectroscopy measurements

Mn K-edge HERFD-XANES spectra were recorded at beamline 15-2 of the Stanford Synchrotron Radiation Lightsource (SSRL) with the SPEC software package v6, while Mn K-edge EXAFS spectra were collected at beamline 9-3 with WEBXAS v2023. At both beamlines, data were collected at 10 K using a liquid He cryostat, and the incident energy was tuned to the first derivative of an internal Mn foil at 6539 eV. X-ray irradiation was carefully monitored so that two subsequent scans of the same spot did not have photoreduction differences, and different spots along the samples were scanned. When appropriate, aluminum foil was inserted into the beam path to attenuate the incident flux. For HERFD-XANES measurements, a Johann-type hard X-ray spectrometer with six Ge(333) analyzer crystals was used with a liquid-nitrogen cooled Si(311) double crystal monochromator, and energy was calibrated to a glitch with measurement of Mn Foil. For EXAFS, measurements were recorded with a 100-element Ge monolithic solid-state fluorescence detector, and a Si(220) monochromator at Φ = 90° was used.

## X-ray absorption spectroscopy data analysis

The post-edge of the XANES was normalized to unity with MATLAB v2023a. Pre-edge peak fitting for HERFD results was performed with pseudo-Voigt functions. Pre-edge intensities are defined as the total trapezoidal numerical integration of the fitted peaks. EXAFS data reduction, averaging, and refinement were carried out with the LARCH software package v0.9.78[97]. Refinement of the $k^2\chi(k)$ EXAFS data used phases and amplitudes obtained from FEFF v8L[98]. For each fit, optimization of the radial distribution around the absorbing Mn ion ($r$) and the Debye-Waller factor ($\sigma^2$) was performed. The goodness-of-fit was evaluated by reduced $\chi^2$ values and R-factors.

## Computational methods

All DFT calculations were performed with the ORCA quantum chemistry package version 5.0 using the B3LYP functional, the def2-TZVP basis set for all atoms, and the CPCM solvation model[99–102]. For geometry optimizations, the full active site (i.e., residues shown in the right panel of Fig. 1a) was included from the neutron structure where the O and N atoms of the peptide backbone were truncated, and $C^\alpha$ was fixed. All atoms for residues Trp161, Phe66, and Tyr166 were fixed with the exception of hydroxyl Tyr166 proton to mimic the packing found in the native enzyme. The Mn ion used the high spin quintet and sextet states for trivalent and divalent systems, respectively, per experimental observations[74]. A dense integration grid and tight convergence were enforced.

For TDDFT calculations, the Mn ion was instead assigned the core property basis set, CP(PPP)[103,104]. The geometry-optimized model was used and truncated to only the Mn ion and its immediate ligands. Inclusion of all active site residues for TDDFT did not significantly alter the simulated spectra. Computed Mn K-pre-edge data were plotted using a Gaussian broadening of 1 eV and a 32.3 eV energy correction was applied in line with previous studies[69,76].

XANES simulations of the Mn K-edge were achieved using the FDMNES quantum chemistry software package v2023[105]. FDMNES uses the finite difference method to solve the Schrödinger-like equations[106]. The scattering potential around the Mn absorber was calculated self-consistently within a radius of 6 Å and used a fully screened core hole. XANES spectra were simulated for Mn complexes derived from the neutron structural data and fit to experimental HERFD-XANES spectra using the Fitit code v2021[73]. For fits of the spectra, the XANES spectra of the neutron structure are first simulated with FDMNES[105]. Systematic deformations of the structure were then applied, and the corresponding spectra of each deformation were simulated to generate a training set. From the training set, each point of a XANES spectrum is defined as a function of structural parameters that are then used for fitting the experimental spectra with structural refinement. Fits were performed by refinement of the Mn bond distances.

## Reporting summary

Further information on research design is available in the Nature Portfolio Reporting Summary linked to this article.

## Data availability

Coordinates and structure factors for neutron and X-ray crystallographic data generated in this study have been deposited in the Protein Data Bank under the following accession codes (8VHW, 8VHY, 8VJ0, 8VJ4, 8VJ5, and 8VJ8). Previously published wildtype neutron structures are available in the Protein Data Bank under the following accession codes (7KKS and 7KKW). X-ray spectroscopy data and coordinates for computational models are provided in the Source Data file. All relevant data supporting the key findings of this study are available within this article, its Supplementary Information, or in the Source data file. Additional raw data is available from the corresponding author upon request. Source data are provided with this paper.

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

## Acknowledgements

This research was supported by the NIH (RO1-GM145647) and NASA EPSCoR (NE–80NSSC17M0030 and NE-NNX15AM82A) grants awarded to G.E.O.B. The UNMC Structural Biology Core Facility was funded by the Fred and Pamela Buffett NCI Cancer Center Support Grant (P30CA036727). The research at Oak Ridge National Laboratory (ORNL) Spallation Neutron Source was sponsored by the Scientific User Facilities Division, Office of Basic Energy Sciences, US Department of Energy. The Office of Biological and Environmental Research supported research at ORNL Center for Structural Molecular Biology (CSMB) using facilities supported by the Scientific User Facilities Division, Office of Basic Energy Sciences, US Department of Energy. Use of the Stanford Synchrotron Radiation Lightsource (SSRL), SLAC National Accelerator

Laboratory, is supported by the US Department of Energy (DOE), Office of Science, Office of Basic Energy Sciences under Contract DE-AC02-76SF00515. The SSRL Structural Molecular Biology Program is supported by the DOE Office of Biological and Environmental Research, and by the National Institutes of Health, National Institute of General Medical Sciences (P30GM133894). The contents of this publication are solely the responsibility of the authors and do not necessarily represent the official views of NIGMS or NIH. Quantum chemical computations were completed using the Holland Computing Center of the University of Nebraska, which receives support from the Nebraska Research Initiative.

## Author contributions

J.A. purified protein, performed X-ray/neutron crystallography and X-ray spectroscopy experiments, processed and analyzed data, performed quantum chemistry calculations, and wrote the manuscript. K.S., L.R.S., and W.E.L. purified protein and participated in data collection and instrument maintenance. L.C., D.A.A.M., and K.W. performed neutron crystallography experiments and processed data. T.K. performed X-ray spectroscopy experiments, processed data, and performed quantum chemistry calculations. G.E.O.B. acquired funding, designed experiments, and participated in writing the manuscript.

## Competing interests

The authors declare no competing interests.
