## [Peer Review File · Nature Communications]

Revealing the atomic and electronic mechanism of human manganese superoxide dismutase product inhibitionREVIEWER COMMENTS

Reviewer #1 (Remarks to the Author):

In this manuscript, the authors present three X-ray and three neutron structures of the W161F mutant of the MnSOD, viz. the reduced and oxidised resting states and the peroxide-inhibited structure. The neutron structures give a detailed picture of the positions of the protons in the active site. The structures are supported by XAS (HERFD and EXAFS) data and DFT calculations. The peroxide-inhibited structure is shown to contain Mn²⁺ with two peroxide molecules bound in one chain, one to Mn, replacing the solvent ligand, and one bridging His30 and Tyr34. Several unusual protonation states are observed, e.g. a deprotonated amide group of Gln143. The results are very interesting and undoubtedly worth publishing. However, many details need to be clarified first.

1. I am a bit surprised about the description of Tyr34 in the previous neutron structures: In the deposited 7kkw structure, Tyr34 in both subunits is deprotonated.
2. k₃+k₄ is exactly the same as k₂, so I cannot see what it has to do with inhibition. They are probably part of the normal reaction mechanism of the enzyme.
3. Differences in the protonation state is determined by the pH, not by the concentration of H₂O₂. All this discussion is very confusing. Inhibition is expected if H₂O₂ binds to Mn²⁺ or more than one H₂O₂ molecule binds (which is actually exactly what is observed in the neutron structures).
4. From which pdb files do the structures in Fig 1h come from? Is the structure of the Trp161Phe mutant already known?
5. The kinetic constants are obtained at a high pH (9.2–9.4 in Table 1), whereas the neutron structure is from pH 7.8. What effects may that have?
6. What is the current view of MnSOD catalysis? Does superoxide bind to the Mn ion or in the second sphere? This should be discussed in the Introduction.
7. The X-ray structures of H₂O₂-soaked MnSOD should be mentioned and described also in the Introduction. They should be compared to the neutron structures in the Results section.
8. What is a “solvent-bound water molecule”?
9. What is the oxidation state of Mn in the D₂O₂-soaked structure. This should be indicated in the title of the section and in Fig. 2. The structure of the other oxidation state would also

be of interest.

10. Is the difference in Gln143-Trp123 H-bond length (0.1 Å) significant at the current resolution?

11. Fig 2c and the text describes only chain A. What are the corresponding distances in chain B?

12. What is the Mn-O(Wat1) bond length in the various structures? There should be a 0.3 Å difference between water and OH⁻, as DFT calculations may confirm.

13. What are the differences in the N(Gln143)-O(Wat1) distances in the corresponding X-ray structures of WT and W161F structures?

14. The authors should be careful deducing binding energies from structures (e.g. for Wat1).

15. It should be mentioned early that the peroxide-inhibited structure is started from mainly the oxidised state of the enzyme but the authors suggest that it is reduced by H₂O₂.

16. The corresponding EXAFS and DFT results from oxidised and reduced enzyme would also be interesting, to get an indication of the accuracy and reliability of the current approach.

17. From kinetic data, what is the predicted % of the inhibited complex for WT and W161F MnSOD?

18. Do the EXAFS data really support fitting of 5 distances and 5 DW factors? Results of fits with different numbers of shells should be given.

19. Can the experiments be repeated for the H₂O₂ inhibited state, started from the reduced state of the metal.

20. Again, regarding the discussion on line 372, DFT results of both HO₂⁻ and HO₂O should be given in Table 2.

21. Also, what are the spin populations on the ligand in the HO₂⁻ and HO₂O complexes?

22. If Mn³⁺ is reduced by H₂O₂ to Mn²⁺ is superoxide detected?

23. What is the nature of the Mn³⁺+HO₂⁻ = Mn²⁺+HO₂O complex (bond lengths and spin populations). Is it pure Mn³⁺ or a mixture? Do the bond lengths agree with the experimental data?

24. Better to write Mn-O bond on line 389.

25. Also Gln143 should be mentioned among the residues with unusual pK_a values.

26. "3a" should be "5a" on line 501.

27. What is the D1-O1-O2-D2 dihedral angle in Fig 5a? H₂O₂ normally has a dihedral of 90–120°.

28. What is the author's definition of a SSHB (based on only structural data)?
29. It sounds very strange that Wat1 would not form a strong H-bond to the negatively charged Gln143 amide HN group. What happens if another proton is added to the amide?
30. Line 593: Better to say "deprotonated" than "unprotonated" as there is still an H atom on the amide N atom.
31. I do not understand the sentence on line 628. Before, it has been said that oxidised MnSOD is reduced by peroxide, but here it is instead suggested that superoxide is the reducing agent. Where does the superoxide come from? In the crystal, 90% of the active sites should be reduced. That is a large amount of superoxide needed.
32. What are the redox potentials of MnSOD, peroxide and superoxide? Is the suggested mechanism feasible from a thermodynamic point of view?
33. The mechanism in Fig 7 does not include the second H₂O₂ molecule observed in the crystal structure. Why not? Is it considered an artifact?
34. L656: "prevent harmful amounts of H₂O₂" – if MnSOD is inhibited, the levels of superoxide will increase, which is even more harmful than peroxide. The level of H₂O₂ should be controlled by catalases.
35. The resolution of the reported crystal structures is relatively low. How much does the limited resolution affect the structure? How certain can we be about the observed structures and differences? A natural suggestion is that the differences observed by chain A and B represent the statistical uncertainty of the structure. This should be discussed.
36. The corresponding X-ray structures are not described at all. What is the resolution of those structures? How well do they agree with the neutron structures, especially in the active site? This should be discussed in detail. A table of Mn-ligand distances should be given for the 6 structures. Do the X-ray structures support the suggested H-bonds, protonations and deprotonations? Are there any indication of photoreduction? Since the (most interesting) peroxide-inhibited structure is Mn²⁺, it should not be not much affected by photoreduction. The same applies to the reduced resting-state structure.
37. Coming back to the reaction mechanisms in the Introduction, k₁ and k₂ are the normal reactions and can only be inhibited by trivial mass action. The same applies to k₃ and k₄. However, binding peroxide to Mn²⁺ is outside the normal reaction cycle and provides a proper mechanism for true inhibition (i.e. formation of a new species). This should be pointed out.

38. The restraints on Phe66, Trp123, Trp161 and Tyr166 in the QM calculations should be specified (which atoms were fixed).

39. What is the concentration of D2O2 in the inhibited structure (in M)?

40. How can a finite difference approach be a DFT method?

Reviewer #2 (Remarks to the Author):

Noteworthy Results and Significance to the Field

- This paper is a tour-de-force that provides urgently-needed detailed chemical insights into mechanisms at work in the active site of a vital metalloenzyme. Critically, by combining EXAFS and XANES with neutron crystallography, the authors unite information on the distributions of valence electrons and the locations of redox-coupled protons. This information is essential to distinguish the substrate and products of the enzyme in question, and to determine how the details of the active site work together to interconvert them, binding one and usually rejecting the other. Indeed, it is the difficulty of rejecting the product from a site evolved to bind the reactant, that can degrade catalytic efficiency. In this case, it is proposed that retention of product may aid in limit turnover and thereby production of the signalling sequellae of product release.

Support for the Conclusions and Methodology

- The quality of the science is excellent, but the writing of the manuscript is currently an impediment to its appreciation.

- Do the 5-coordinate vs. 6-coordinate states found agree with the determinations of Valentine based on EPR?

Mechanism Proposed

- Regarding the mechanism proposed at the end, please consider the following possibility: Inhibition results from interaction with both a molecule of superoxide and one of H2O2, for example as could occur at high superoxide concentrations when an incoming superoxide might arrive before full departure of a recently formed peroxide. Peroxide is formed along with the Mn³⁺ state, and the latter would be reduced to Mn²⁺ by incoming superoxide. However the still-present H2O2 could be deprotonated by the coordinated OH⁻ which

abstracts a proton to form coordinated water in conjunction with reduction of Mn. This is consistent with the lower pH of H₂O₂ than nascent H₂O. Anionic HO₂⁻ could then displace neutral H₂O from its axial coordination site to yield the pentacoordinate Mn²⁺ with axial HO₂⁻ proposed by this work. The 'departing product' would be O₂ in the above scenario, explaining why it could escape trapping, being replaced instead by a second H₂O₂. Just an idea, please consider in particular the first half of it.

Suggestions for readability and clarity:

- I have provided numerous suggestions aimed at streamlining wording and avoiding vocabulary that is off the mark or distracting. I highly recommend that if the authors do not like my suggestions, they should engage a third party to refine.

My proposed changes to wording are in red.

Additional comments (not to be incorporated) are in purple.

- There is some confusion due to initial definition of 'MnSOD' to mean the MnSOD of human mitochondria, specifically. Could 'huMnSOD' be used for this purpose, to leave 'MnSOD' for wider use?

- Use of the yellow/orange superposition in some figures was not effective. Please try green/orange, or orange/blue.

- A 'metal' is a material whose atoms have formal oxidation states of zero, usually in this paper I believe 'metal ion' is intended.

- The authors refer to a protonated Gln side chain.

Please use 'neutral' instead, because a protonated (positively charged) Gln⁺ is also possible and implied by IR measurements in a different system.

- Please clarify if the numerous H-bond lengths quoted are from acceptor heavy atom to H atom, or distances between heavy atoms. Most protein biophysicists will assume the latter.

Reviewer #3 (Remarks to the Author):

Azadmanesh et al. report a study that uses a combination of neutron diffraction, XAS spectroscopy and DFT calculations to understand the molecular basis of the product inhibition mechanism of manganese superoxide dismutase (MnSOD). MnSODs are essential

enzymes that use the Mn³⁺/Mn²⁺ couple to convert the superoxide anion free radicals (O₂^{•-}) to hydrogen peroxide (H₂O₂) in a proton-dependent manner. Human MnSOD is known to be subject to feedback inhibition by the product due to the low H₂O₂-tolerance of mitochondria, although the molecular mechanism of such a feedback inhibition is not well understood. In this study, the authors aimed to investigate the molecular mechanism of product inhibition of MnSOD through the structural characterization of a W161F variant of this enzyme, which was previously shown to be biased towards the product inhibited pathway. By solving the neutron structures of W161F MnSOD in different states and in the presence of D₂O₂, the authors showed that the Mn active site adopted a 5-coordinate Mn²⁺ form both in the resting state and upon peroxide treatment, an assignment further verified by HERFD-XAS spectroscopy. Moreover, they observed that an OOH moiety—proposed to be a hydroperoxyl anion (HO₂⁻) on the basis of XAS and DFT studies—was bound to Mn while forming an H-bond to Q143 in the D₂O₂-treated W161F variant, with a second H₂O₂ moiety at a distal site forming H-bonds with H30 and Y34. Notably, the authors revealed that the distal H₂O₂ moiety formed an extensive H-bonding network that linked Q143 and the Mn active site, and that such an H-bonding network was absent in the resting state where the peroxide/hydroperoxyl anion binding sites were occupied by water molecules.

This is a well-conducted study that explores the mechanism of a significant process that regulates the activity of SOD and the intracellular level of H₂O₂. The authors used an excellent combination of structural, spectroscopic, and theoretical approaches to provide mutually supportive evidence in support of their conclusions; in particular, the XAS analysis that accompanied the neutron diffraction experiments was quality work that facilitated some of the key interpretations of the structural data, such as the assignment of a hydroperoxyl anion at the Mn active site. Overall, the quantity and quality of the data presented by the authors are outstanding, and the implications of their findings are more than intriguing with respect to both the inner workings of MnSODs and the PCET process in general. Because of these reasons, this work warrants publication in *Nature Communications*.

There are, however, several key points that the authors should consider addressing prior to the publication of this manuscript:

(1) Based on their observation of a second, distal peroxide moiety, the authors suggest that product inhibition could be initiated by H₂O₂ that has already been formed instead of directly by the product that was on-path to its exit. This is a significant finding; however, it is unclear whether the extra peroxide is physiologically relevant or if it is an artifact resulting from the high concentration of H₂O₂ used in the experiments. The authors should consider including some relevant kinetic data to strengthen their argument, as product inhibition could very well occur at an exogenous H₂O₂ concentration that is within the range of the reactivity of MnSOD.

(2) It is unclear whether the state generated by soaking with H₂O₂ faithfully represents a naturally occurring inhibited state due to an accumulation of the product. Can this effect be verified by comparing the kinetic or spectroscopic features of the H₂O₂ treated sample with the substrate turnover sample? Would prolonged reactions and accumulation of products, like those occurring in the cell, lead to similar structures or states observed for the H₂O₂ soaked enzyme?

(3) It appears that the extensive hydrogen bonding network observed in the peroxide bound structure (Fig. 6a) could stabilize the intermediate-bound active site and thereby stall the reaction. This possibility could be tested by mutating H30 and Y34 or performing preliminary MD calculations that compare the energetics of the peroxide bound state (Fig. 6a) with those of the other states.

A few minor points for the authors' consideration:

a. The background section, particular with respect to the kinetic features of product inhibition, does not provide adequate information for a layman. The paragraph from ln 91 to ln 100 should be expanded given the crucial importance of the phenomenon described herein for this study.

b. In Figs 2 and 5, the authors jump between chain A and chain B across the different states (D₂O₂ soaked, reduced, and oxidized states) of the enzyme. For the purpose of a better comparison, the authors should include a parallel illustration of chain A and B for all 3

reported states in the supplemental information.

c. Fig. 6 is a great figure summarizing all the findings in this manuscript. However, it would be helpful for the readers if the authors could cross reference the structures shown in this figure with the various experiments described in the manuscript.

Reviewer #4 (Remarks to the Author):

Azadmanesh and co-authors have determined neutron structures of human manganese superoxide dismutase (MnSOD) with/without D₂O₂ soaking. Neutron structures elucidated the protonation states of amino acids and the hydrogen bond network in the D₂O₂-soaked, reduced, and oxidized MnSOD W161F mutant. The comparison between these structures, including MnSOD WT structures, allows us to understand the PCET catalysis of MnSOD. In addition, XAS and QM (DFT) calculations correlated well with neutron structures and complemented them. Therefore, the proposed atomic and electronic mechanism for MnSOD product inhibition and relief should be supported. On the other hand, the hydrogen bond information, especially SSHB, is unreliable because the coordinates of many hydrogens (deuteriums) are shifted from the HD-omit neutron maps. The authors should mention these deviations and, if possible, provide coordinates and structure factors to the reviewers.

Other points

1. As the authors mentioned, neutron crystallography gives us structure information without radiation-induced artifacts. Do the authors consider the X-ray structure information shown in supplementary Fig. 1 to be an artifact? If so, the authors should exclude X-ray structure analyses from the manuscript. (The supplementary Fig. 1 will only confuse the readers).
2. The authors used W161F mutant to trap the product-inhibited complex. Have they tried using wild type? By controlling the soaking times, the oxidized resting state and inhibitor complex may be distinguished. If so, this manuscript will be even more substantial.
3. The authors should add bond lengths from DFT calculation using a hydroperoxy radical (HO₂[·]) to Table 2.
4. The authors should show unoccupied residual density by calculating D₂O, not D₂O₂, as a

supplementary figure.

Minor points

1. line 501: Fig. 3a → Fig. 5a2.
2. Which crystals were used for neutron structure analyses, obtained in space or on the ground?
3. The $CC_{1/2}$ values in the outer shell of W161F reduced and oxidized neutron structures below 0.5. Are these d_{\min} adequate?

REVIEWER COMMENTS

Reviewer #1 (Remarks to the Author):

We kindly thank the reviewers for their enthusiasm for our research and the thorough evaluation of the manuscript. The suggestions have undoubtedly strengthened the manuscript and improved the quality and presentation of the work.

In this manuscript, the authors present three X-ray and three neutron structures of the W161F mutant of the MnSOD, viz. the reduced and oxidized resting states and the peroxide-inhibited structure. The neutron structures give a detailed picture of the positions of the protons in the active site. The structures are supported by XAS (HERFD and EXAFS) data and DFT calculations. The peroxide-inhibited structure is shown to contain Mn²⁺ with two peroxide molecules bound in one chain, one to Mn, replacing the solvent ligand, and one bridging His30 and Tyr34. Several unusual protonation states are observed, e.g. a deprotonated amide group of Gln143. The results are very interesting and undoubtedly worth publishing. However, many details need to be clarified first.

1. I am a bit surprised about the description of Tyr34 in the previous neutron structures: In the deposited 7kkw structure, Tyr34 in both subunits is deprotonated.

Thank you for pointing this out! Unbeknownst to us, four deuterium atoms were removed during the PDB deposition process, including the proton of Tyr34 in chain B. These atoms used unconventional PDB syntax and were removed during the conversion to the mmCIF coordinate file format. We have corresponded with the PDB and have submitted a correction. Please note the density for the Tyr34 proton was published in our 2021 Nature Communications publication in Fig 4b¹. Should the reviewer request it, we will provide the corrected PDB file for entry 7KKW.

2. k₃+k₄ is exactly the same as k₂, so I cannot see what it has to do with inhibition. They are probably part of the normal reaction mechanism of the enzyme.

While the chemical equations of k₃ and k₄ added together resemble that of k₂, the zero-order process of k₄ means that O₂^{•-} is not reacting with MnSOD during this phase, and the enzyme is “product inhibited.”

We have expanded the introductions in an effort to make this more clear in lines 110-125.

3. Differences in the protonation state is determined by the pH, not by the concentration of H₂O₂. All this discussion is very confusing. Inhibition is expected if H₂O₂ binds to Mn²⁺ or more than one H₂O₂ molecule binds (which is actually exactly what is observed in the neutron structures).

Yes, differences in protonation are indeed affected by pH! However, enzymes that use proton transfers to facilitate catalysis are expected to undergo changes in active site protonation states based on the presence of substrate or product. This is observed in several other neutron structures where the protonation states alter depending on the ligand bound²⁻⁴. We have added this clarification in the first paragraph of section “The structural identity of the product-inhibited complex” in lines 180-183.

4. From which PDB files do the structures in Fig 1h come from? Is the structure of the Trp161Phe mutant already known?

The wildtype X-ray structure is from our previous work, PDB ID 5VF9. The Trp161Phe X-ray structure is from this work, PDB ID 8VJ8. We have included the PDB IDs in the Fig. 1 legend and in Supplementary Table 5.

5. The kinetic constants are obtained at a high pH (9.2–9.4 in Table 1), whereas the neutron structure is from pH 7.8. What effects may that have?

We found a reference with measured values at pH 8.2, and the values are nearly identical to pH 9.2–9.4⁵. We altered Table 1 to use the values in this reference since the pH is much closer to our structures.

The pH of our crystallization process (pH 7.8) physiological pH of mitochondria and best reflects the protonation states that would occur in cells. These details have been added to the first paragraph of the results section.

6. What is the current view of MnSOD catalysis? Does superoxide bind to the Mn ion or in the second sphere? This should be discussed in the Introduction.

As suggested, we have added a discussion of how superoxide interacts with the active site to the introduction in lines 92-99.

7. The X-ray structures of H₂O₂-soaked MnSOD should be mentioned and described also in the Introduction. They should be compared to the neutron structures in the Results section.

The X-ray structures are found in Supplementary Fig. 1 along with the accompanying bond distances in Supplementary Table 1. The dioxygen species bound to the Mn ion in these structures are in different orientation and are partially occupied due to the effects of photoreduction. Note that X-ray irradiation affects not only the metal oxidation state but also brings the effects of radiolysis on water and other components of the solvent (e.g. H₂O₂)⁶⁻⁸. These effects complicate X-ray structural analysis of metalloenzymes, especially with metal-bound dioxygen species⁹. We note several other PDB structures of metalloenzymes that have dioxygen species with partial occupancy or appreciably high B factors compared to the rest of the protein (7CIT, 7KQU, 6LQW, 6K0F, 6K0E, 7CIY, 6LF7, 7DN7, 6LRN, 7DN6, 7DLQ).

Reviewer #4 suggested removing X-ray structural analysis from the manuscript since these artefacts may confuse reviewers. We feel including the Supplementary Fig. 1 strengthens the rationale for pursuing neutron crystallography and bolsters the overall impact of the work. We have described both in the main manuscript and the Supplementary Figure legend that the X-ray structures are influenced by photoreduction effects. We also altered the sentence following the first mention of Supplementary Fig. 1 to read “The dioxygen molecules in these X-ray structures are different in orientation and partially occupied due to X-ray irradiation and demonstrate the benefit of using neutrons for radiation-free structural analysis.” We also added the occupancy numbers to Supplementary Fig. 1 to accentuate the concept.

8. What is a “solvent-bound water molecule”?

A mistake on our end – we have corrected it to “metal-bound water molecule.”

9. What is the oxidation state of Mn in the D₂O₂-soaked structure. This should be indicated in the title of the section and in Fig. 2. The structure of the other oxidation state would also be of interest.

Our XAS data indicate that the oxidation state is Mn²⁺. However, because we have not brought up XAS at this point in the manuscript, we have refrained from identifying it. We added a sentence to the end of the section: "The valency of the Mn ion and identity of the dioxygen species will be explored in later sections."

We were actually expecting D₂O₂-soaking to lead to a [Mn³⁺-OOH] species, and we were surprised that with treatment with peroxide led to an Mn²⁺ species. Previous literature postulated the product-inhibited complex to be Mn³⁺ containing^{10,11}. To our knowledge, there is no experimental evidence of such a complex in MnSOD.

10. Is the difference in Gln143-Trp123 H-bond length (0.1 Å) significant at the current resolution?

We presume this is in regard to Fig. 2f. This is not significant at our current resolution. We have removed the discussion of this difference from the text. We have also added the coordinate error calculated by PHENIX to Supplementary Table 5.

11. Fig 2c and the text describes only chain A. What are the corresponding distances in chain B?

For the resting state structures, only one active site is shown due to high structural similarities. The corresponding distances in chain B are now shown in Supplementary Fig. 2. These details have been added to the Fig. 2 legend.

12. What is the Mn-O(Wat1) bond length in the various structures? There should be a 0.3 Å difference between water and OH-, as DFT calculations may confirm.

Yes, there is a difference. We have included the Mn-O(WAT1) bond lengths in Supplementary Table 1. We have also included comparisons with DFT in Supplementary Table 3.

13. What are the differences in the N(Gln143)-O(Wat1) distances in the corresponding X-ray structures of WT and W161F structures?

The wildtype and W161F structures have distances of 2.8 and 3.1 Å, respectively. This is now denoted in Fig. 1h.

14. The authors should be careful deducing binding energies from structures (e.g. for Wat1).

We presume this is in reference to the sentence 'Compared to HO₂[•], HO₂⁻ is a more attractive candidate to displace the WAT1 ligand.' We removed this sentence!

15. It should be mentioned early that the peroxide-inhibited structure is started from mainly the oxidised state of the enzyme but the authors suggest that it is reduced by H₂O₂.

We have now added this to the introduction lines 141-163.

16. The corresponding EXAFS and DFT results from oxidised and reduced enzyme would also be interesting, to get an indication of the accuracy and reliability of the current approach.

The corresponding DFT results and XANES fits are found in Supplementary Table 3. Unfortunately, due to a lack of beamtime, we do not have EXAFS data on the corresponding oxidized and reduced enzyme, though this would surely be of benefit! Please note that we used a multitude of approaches to study the Mn-ligand bond distances of the H₂O₂-soaked enzyme: neutron crystallography, DFT, EXAFS fitting, and XANES fitting. We also provide the distances of the oxidized and reduced counterparts from the approaches of neutron crystallography, DFT, and XANES fitting.

17. From kinetic data, what is the predicted % of the inhibited complex for WT and W161F MnSOD?

This can be roughly estimated by comparing the rates of k_2 (reaction to form H₂O₂) and k_3 (formation of the inhibited complex). However, these estimates assume that $[O_2^{\bullet-}] \gg [MnSOD]$, that all MnSOD start from the Mn²⁺ oxidation state, and that the effects of k_1 and k_4 are neglected.

For wildtype, since k_2 and k_3 have identical magnitudes, they are competing reactions, and a simplified estimate would be 50% inhibited. For Trp161Phe, because k_2 is miniscule, leading to $k_2 \ll k_3$, the enzyme would be ~99% inhibited.

We expanded on the relationship between k_2 and k_3 in the introduction and noted the percent inhibition with caveats lines 110-125.

18. Do the EXAFS data really support fitting of 5 distances and 5 DW factors? Results of fits with different numbers of shells should be given.

In addition to the five-coordinate fit, we added six-coordinate and four-coordinate fits to Supplementary Table 2. The fitting results for the five-coordinate complex best fit the data.

19. Can the experiments be repeated for the H₂O₂ inhibited state, started from the reduced state of the metal.

This is a great question that we have wondered ourselves and would like to explore with future beamtimes. To our knowledge, there is no evidence in the literature that Mn²⁺SOD and H₂O₂ forms the inhibited complex. Rather, the literature supports forming the inhibited complex through soaking H₂O₂ with Mn³⁺SOD^{5,12,13}.

20. Again, regarding the discussion on line 372, DFT results of both HO₂⁻ and HO₂O should be given in Table 2.

The corresponding bond lengths have been added.

21. Also, what are the spin populations on the ligand in the HO₂⁻ and HO₂O complexes?

The spin populations have been added to Supplementary Table 4.

22. If Mn³⁺ is reduced by H₂O₂ to Mn²⁺ is superoxide detected?

This is a good question! Previous scanning stopped-flow spectrophotometry studies by Hearn and colleagues noted that mixing H_2O_2 and Mn^{3+}SOD led to rapid formation of the inhibited complex, though observation of a $\text{O}_2^{\cdot-}$ complex is lost during the dead time of the instrument (1.4 ms)¹³. These details have been added in lines 389-392.

23. What is the nature of the $\text{Mn}^{3+}\text{HO}_2^- = \text{Mn}^{2+}\text{HO}_2\text{O}$ complex (bond lengths and spin populations). Is it pure Mn^{3+} or a mixture? Do the bond lengths agree with the experimental data?

We have added the corresponding DFT bond distances to Table 2 and DFT spin populations to Supplementary Table 4. They compare well with previous MnSOD DFT calculations¹⁴.

Attempts to geometry optimize an antiferromagnetically coupled ($S=2$) $[\text{Mn}^{2+}\cdot\text{OOH}]$ with spin-flip broken-symmetry DFT collapses to a state where $\rho_s(\text{Mn}) = 4.03$ and $\rho_s(\text{O}_2) = -0.07$. This indicates a complex mostly of the $[\text{Mn}^{3+}\cdot\text{OOH}]$ state.

Geometry optimizing ferromagnetically coupled ($S=3$) $[\text{Mn}^{2+}\cdot\text{OOH}]$ leads to $\rho_s(\text{Mn}) = 4.86$ and $\rho_s(\text{O}_2) = 0.85$. This corresponds to a complex mostly of the $[\text{Mn}^{2+}\cdot\text{OOH}]$ state.

A single point broken-symmetry $S = 2$ calculation with the geometry found with $S = 3$ leads to $\rho_s(\text{Mn}) = 4.73$ and $\rho_s(\text{O}_2) = -0.81$. This approach was previously used with MnSOD DFT calculations to obtain negative spin densities for the dioxygen species since geometry optimization would lead to a collapse of the broken-symmetry spin state¹⁴.

The bond lengths of the experimental data still best correspond to $[\text{Mn}^{2+}\cdot\text{OOH}]$.

24. Better to write $\text{Mn}-\text{O}$ bond on line 389.

We have altered the text as suggested!

25. Also Gln143 should be mentioned among the residues with unusual pKa values.

We have also added Gln143!

26. "3a" should be "5a" on line 501.

Thank you, we have altered the text as suggested.

27. What is the $\text{D1}-\text{O1}-\text{O2}-\text{D2}$ dihedral angle in Fig 5a? H_2O_2 normally has a dihedral of $90-120^\circ$.

The dihedral angle is 17° . Both D atoms of D_2O_2 are within hydrogen bond distance of anionic Tyr34 and may explain the unusual angle the neutron diffraction data indicates. We have added these details to the manuscript in the section "Unexpected second-sphere hydrogen peroxide binding site" lines 536-537.

28. What is the author's definition of a SSHB (based on only structural data)?

From a structural point of view, we are defining SSHBs as hydrogen bonds that are less than 1.8 \AA . This is stated in the introduction, lines 65-66.

29. It sounds very strange that Wat1 would not form a strong H-bond to the negatively charged Gln143 amide HN group. What happens if another proton is added to the amide?

Yes, this is indeed unusual. If we add the other proton to the Gln to achieve the neutral state during the model building and refinement process, negative $F_{\text{O}}-F_{\text{C}}$ neutron scattering length density appears, indicating this is in error. Additionally, this would result in a steric collision with the $D^2(\text{WAT1})$ proton as the interaction distance between the two D atoms would be less than 1.5 Å. Note that we are careful during our model building and refinement to place the most chemically sensible arrangement of atoms from the nuclear density.

30. Line 593: Better to say “deprotonated” than “unprotonated” as there is still on H atom on the amide N atom.

We have altered the text as suggested!

31. I do not understand the sentence on line 628. Before, it has been said that oxidised MnSOD is reduced by peroxide, but here it is instead suggested that superoxide is the reducing agent. Where does the superoxide come from? In the crystal, 90% of the active sites should be reduced. That is a large amount of superoxide needed.

We have rewritten this section!

The approximate molar ratio of MnSOD to $\text{H}_2\text{O}_2/\text{D}_2\text{O}_2$ is 1:100. In our experiments, we believe that mixing Mn^{3+}SOD with excessive concentrations of $\text{H}_2\text{O}_2/\text{D}_2\text{O}_2$ triggers a backwards reaction to produce Mn^{2+}SOD and $\text{O}_2^{\bullet-}$. This would be the most likely explanation as to why H_2O_2 reduces Mn^{3+}SOD .

From our experiments using high concentrations of $\text{H}_2\text{O}_2/\text{D}_2\text{O}_2$, we postulate that the inhibited complex is from first a backwards reaction of H_2O_2 with Mn^{3+} to form a $\text{O}_2^{\bullet-}$ species and Mn^{2+} , and then a forward reaction of $\text{O}_2^{\bullet-}$ with another Mn^{3+} site in the presence of H_2O_2 to form the inhibited complex. Here, H_2O_2 is seen as a means to generate $\text{O}_2^{\bullet-}$ and study the forward reactions, and we have framed our proposed mechanism only in the forward direction to better represent what would occur in physiological conditions.

32. What are the redox potentials of MnSOD, peroxide and superoxide? Is the suggested mechanism feasible from a thermodynamic point of view?

The reduction potential of wildtype Mn^{3+}SOD to Mn^{2+}SOD (0.4 V) is between the oxidation of $\text{O}_2^{\bullet-}$ to O_2 (-0.16 V) and the reduction of $\text{O}_2^{\bullet-}$ to H_2O_2 (0.85 V)¹⁵.

When calculating whether Mn^{3+}SOD being reduced by H_2O_2 is thermodynamically favorable – adding the reduction potential of Mn^{3+}SOD (0.4 V) and the oxidation potential of H_2O_2 (-0.85 V), the resulting value is negative to indicate the process is not thermodynamically favorable.

However, Bull *et al.* and Hearn *et al.* have noted that mixing Mn^{3+}SOD and H_2O_2 leads to a slow reaction yielding Mn^{2+}SOD ^{13,16}. The process was best fit by second-order kinetics, where reduction of Mn^{3+}SOD and H_2O_2 results in Mn^{2+}SOD and $\text{O}_2^{\bullet-}$ (a back reaction), and then a subsequent reaction of $\text{O}_2^{\bullet-}$ with a second Mn^{3+}SOD to yield another Mn^{2+}SOD site (a forward reaction). Several studies have further observed that introducing high concentrations of H_2O_2 to Mn^{3+}SOD leads to formation of the

inhibited complex that decays to $Mn^{2+}SOD^{5,12,13,16}$. These details have been added to the discussion portion of the section “The electronic identity of the product-inhibited complex” in lines 378-385.

The high concentrations of H_2O_2 that we soak the enzyme in are not possible at physiological conditions. Instead, we think of H_2O_2 as a means to generate $O_2^{\bullet-}$ through a backward reaction and then instigate forward reaction to form the inhibited complex. This is now discussed in lines 645-649 as well as in the introduction in lines 155-163.

Along with a suggestion made by reviewer #2, we have reframed our suggested mechanism to be relevant only to the forward reaction direction. Product inhibition is dependent on a molecule of $O_2^{\bullet-}$ arriving and reducing Mn^{3+} before the departure of a recently formed H_2O_2 . This now places the mechanism in the realm of thermodynamic favorability (since it only corresponds to the forward reaction direction) and better represents what may occur in physiological conditions.

33. The mechanism in Fig 7 does not include the second H_2O_2 molecule observed in the crystal structure. Why not? Is it considered an artifact?

The second D_2O_2 molecule is in Fig. 7a, just above the Mn and to the left of Tyr34!

34. L656: “prevent harmful amounts of H_2O_2 ” – if MnSOD is inhibited, the levels of superoxide will increase, which is even more harmful than peroxide. The level of H_2O_2 should be controlled by catalases.

We have rewrote the paragraph of the line mentioned to more carefully phrase the role of H_2O_2 in cellular dysfunction. We now say “helps prevent cellular dysfunction” rather than “prevent harmful amounts of H_2O_2 ” in line 674.

Most cells do not have mitochondrial catalase^{17,18}. Limiting H_2O_2 formation may be as crucial as the rate of superoxide removal since H_2O_2 is freely diffusible across membranes, and superoxide is not. It is important to note that H_2O_2 produced from MnSOD also serves as a signaling molecule with a wide variety of cellular effectors^{18,19}. These details have been added in lines 100-109.

Expression of MnSOD mutants that have deficient product-inhibition leads to elevated cellular ROS levels compared to wildtype enzyme²⁰. We have included these details in the introduction in lines 105-107.

35. The resolution of the reported crystal structures is relatively low. How much does the limited resolution affect the structure? How certain can we be about the observed structures and differences? A natural suggestion is that the differences observed by chain A and B represent the statistical uncertainty of the structure. This should be discussed.

We now include parallel figures of both chains from the neutron structures in Supplementary Fig. 2. Overall, the chains of the resting state are similar and do not change the conclusions of the mechanism. For the D_2O_2 -soaked structure, only one chain has a dioxygen species bound, and the obvious differences between its chains gives us confidence in the presence of the dioxygen species. Note that at resolutions $< 2.5 \text{ \AA}$, we are confident is seeing hydrogen(deuterium) positions.

Additionally, we have now incorporated coordinate error statistics into Supplementary Table 5. Note that the disorder of atom positions is reflected by the *B*-factor.

36. The corresponding X-ray structures are not described at all. What is the resolution of those structures? How well do they agree with the neutron structures, especially in the active site? This should be discussed in detail. A table of Mn-ligand distances should be given for the 6 structures. Do the X-ray structures support the suggested H-bonds, protonations and deprotonations? Are there any indication of photoreduction? Since the (most interesting) peroxide-inhibited structure is Mn²⁺, it should not be not much affected by photoreduction. The same applies to the reduced resting-state structure.

The H₂O₂-soaked X-ray structures are found in Supplementary Fig. 1 along with H-bond distances. The X-ray Trp161Phe structure is now found in Fig. 1h along with H-bond distances. The Mn-ligand bond distances for all structures are now given in Supplementary Table 1. Resolutions are now given in the accompanying figure legends along with Supplementary Table 5. Overall, the X-ray structures of H₂O₂ soaked and reduced resting do not agree with the neutron. This is why we pursued several different methods to identify the bond distances – DFT and XANES fitting.

For the peroxide-inhibited X-ray structures, the dioxygen species bound to the Mn ion in these structures are in a different orientation and are partially occupied due to the effects of photoreduction. As a result, the Mn-ligand bond distances and H-bonds differ from those found by neutrons.

Structural analysis of dioxygen species bound to the active site of metalloproteins is particularly challenging with X-rays⁹, as noted by several other PDB structures that have dioxygen species with partial occupancy or appreciably high B factors compared to the rest of the protein (7CIT, 7KQU, 6LQW, 6K0F, 6K0E, 7CIY, 6LF7, 7DN7, 6LRN, 7DN6, 7DLQ). We altered the sentence following the first mention of Supplementary Fig. 1 to note the dioxygen molecules in these X-ray structures are different in orientation and are partially occupied due to the effects of X-ray irradiation. This demonstrates the benefit of using neutrons for photoreduction-free structural analysis. Additionally, we added the occupancy numbers to Supplementary Fig. 1 to accentuate the concept.

For the reduced resting state X-ray structures, the Mn-ligand bond does not agree with that of the neutron (Supplementary Table 1). This is the case for both our wildtype and Trp161Phe neutron structures. Additionally, the H-bond distances do not agree between X-ray (Fig. 1h) and neutron (Fig. 2c).

Overall, we attribute the discrepancies of the Mn²⁺-containing structures to X-ray damage of dioxygen species and the solvent molecule bound to the metal ion. Solvent molecules and dioxygen species are susceptible to radiolysis, even at 77K where electrons have been shown to be mobile⁶⁻⁸. Indeed, our solvent structure and hydrogen bonds are persistently different between X-ray/neutron pairs – even from the same crystal (7KKW and 7KLB, neutron data were collected first then the X-ray data from the same crystal)¹. Because the metal-bound solvent and dioxygen species are central to the hydrogen bonding network, it is sensible that damage to these molecules would alter hydrogen bond distances.

As a result of these discrepancies, we refrain from discussing the X-ray structures in detail. Additionally, the main manuscript is already dense with data analysis and discussion, and we feel that incorporating further discussion of the X-ray structures would stray away from the main goal of our work – determining the protonation states of the active site and the electronic state of the Mn ion.

37. Coming back to the reaction mechanisms in the Introduction, k₁ and k₂ are the normal reactions and can only be inhibited by trivial mass action. The same applies to k₃ and k₄. However, binding

peroxide to Mn²⁺ is outside the normal reaction cycle and provides a proper mechanism for true inhibition (i.e. formation of a new species). This should be pointed out.

Thank you for the suggestion – we have expanded the introduction to include these details of k₁-k₄ in lines 110-125.

38. The restraints on Phe66, Trp123, Trp161 and Tyr166 in the QM calculations should be specified (which atoms were fixed).

We have edited the text accordingly in lines 801-803.

39. What is the concentration of D₂O₂ in the inhibited structure (in M)?

The final concentration of D₂O₂ was 280 mM for the solution the crystal was immersed in prior to data collection. This detail has been added to the methods section 'Crystal Manipulations' in line 739.

40. How can a finite difference approach be a DFT method?

The FDMNES software package uses a finite difference method to solve the Schrödinger-like equations for QM simulations²¹. These details have been added to the methods section "Computational Methods," last paragraph, in line 812.

To avoid confusion, we also have removed mention of DFT when describing FDMNES and its corresponding simulations.

Reviewer #2 (Remarks to the Author):

Noteworthy Results and Significance to the Field

- This paper is a tour-de-force that provides urgently-needed detailed chemical insights into mechanisms at work in the active site of a vital metalloenzyme. Critically, by combining EXAFS and XANES with neutron crystallography, the authors unite information on the distributions of valence electrons and the locations of redox-coupled protons. This information is essential to distinguish the substrate and products of the enzyme in question, and to determine how the details of the active site work together to interconvert them, binding one and usually rejecting the other. Indeed, it is the difficulty of rejecting the product from a site evolved to bind the reactant, that can degrade catalytic efficiency. In this case, it is proposed that retention of product may aid in limit turnover and thereby production of the signalling sequellae of product release.

Support for the Conclusions and Methodology

- The quality of the science is excellent, but the writing of the manuscript is currently an impediment to its appreciation.

We thank the reviewer for appreciating the science, suggestions to make the manuscript clearer, and suggestions that improved our mechanistic model.

- Do the 5-coordinate vs. 6-coordinate states found agree with the determinations of Valentine based on EPR?

We presume this is in reference to the elegant study published in JACS by the Valentine group "A Comparison of Two Yeast MnSODs: Mitochondrial *Saccharomyces cerevisiae* versus Cytosolic *Candida albicans*." In this work, EPR spectra support the presence of a six-coordinated Mn^{3+} SOD complex from *S. cerevisiae*²². Our neutron and XAS data of human wildtype and Trp161Phe Mn^{3+} SOD indicate a five-coordinate complex. It is, however, interesting that differences are observed in kinetics, level of product-inhibition, and preferred resting oxidation states between ScMnSOD and huMnSOD. We implemented the details of the ScMnSOD EPR results into the last paragraph of the section "The structural identity of the product-inhibited complex" in lines 213-215.

Mechanism Proposed

- Regarding the mechanism proposed at the end, please consider the following possibility: Inhibition results from interaction with both a molecule of superoxide and one of H_2O_2 , for example as could occur at high superoxide concentrations when an incoming superoxide might arrive before full departure of a recently formed peroxide. Peroxide is formed along with the Mn^{3+} state, and the latter would be reduced to Mn^{2+} by incoming superoxide. However the still-present H_2O_2 could be deprotonated by the coordinated OH^- which abstracts a proton to form coordinated water in conjunction with reduction of Mn. This is consistent with the lower pH of H_2O_2 than nascent H_2O . Anionic HO_2^- could then displace neutral H_2O from its axial coordination site to yield the pentacoordinate Mn^{2+} with axial HO_2^- proposed by this work. The 'departing product' would be O_2 in the above scenario, explaining why it could escape trapping, being replaced instead by a second H_2O_2 . Just an idea, please consider in particular the first half of it.

This is a great suggestion, and we have decided to implement the first half of it. We rewrote the proposed mechanism section to incorporate the idea that $O_2^{\bullet-}$ arrives and reduces Mn^{3+} before the departure of a recently formed H_2O_2 . However, our structures better support H_2O_2 donating a proton to Tyr34 rather than the coordinated $^{\bullet}OH$ molecule, as the two are engaging in strong hydrogen bond interactions and Tyr34 is shown to have different protonation states (Fig. 5). Our structures do not show evidence of an H_2O_2 to $^{\bullet}OH$ proton transfer (i.e. the two are not shown to be hydrogen bonding).

Suggestions for readability and clarity:

- I have provided numerous suggestions aimed at streamlining wording and avoiding vocabulary that is off the mark or distracting. I highly recommend that if the authors do not like my suggestions, they should engage a third party to refine.

My proposed changes to wording are in red.

Additional comments (not to be incorporated) are in purple.

We have not received an attachment detailing these suggestions. However, we have made a large effort to streamline and improve the writing of the manuscript. Please note that we expanded the introduction in response to other reviewers to better explain MnSOD kinetics and the unusual effects in the data, such as H_2O_2 acting as a reducing agent.

- There is some confusion due to initial definition of 'MnSOD' to mean the MnSOD of human mitochondria, specifically. Could 'huMnSOD' be used for this purpose, to leave 'MnSOD' for wider use?

This is an idea we wrestled with! We believe that leaving it as MnSOD is most appropriate since our study is focused entirely on human MnSOD. We introduce the MnSOD under study as human in both the title and first sentence of the abstract, and we discuss MnSOD in relevance to human disease in the introduction. Additionally, leaving the terminology as 'MnSOD' would support a more streamlined version of the manuscript, as suggested above.

- Use of the yellow/orange superposition in some figures was not effective. Please try green/orange, or orange/blue.

We have altered the superposition towards an orange/blue color scheme, as suggested.

- A 'metal' is a material whose atoms have formal oxidation states of zero, usually in this paper I believe 'metal ion' is intended.

We have changed the text to 'metal ion' where appropriate.

- The authors refer to a protonated Gln side chain. Please use 'neutral' instead, because a protonated (positively charged) Gln⁺ is also possible and implied by IR measurements in a different system.

We have altered the text as suggested.

- Please clarify if the numerous H-bond lengths quoted are from acceptor heavy atom to H atom, or distances between heavy atoms. Most protein biophysicists will assume the latter.

At the first mention of neutron structure hydrogen bond distances in the text (lines 65-66), and in the Figure 1 legend, we now clarify that the distances are between donor D/H atom and acceptor atom for the neutron structures. For X-ray structures they are between nitrogen and oxygen atoms.

Reviewer #3 (Remarks to the Author):

Azadmanesh et al. report a study that uses a combination of neutron diffraction, XAS spectroscopy and DFT calculations to understand the molecular basis of the product inhibition mechanism of manganese superoxide dismutase (MnSOD). MnSODs are essential enzymes that use the Mn³⁺/Mn²⁺ couple to convert the superoxide anion free radicals (O₂^{•-}) to hydrogen peroxide (H₂O₂) in a proton-dependent manner. Human MnSOD is known to be subject to feedback inhibition by the product due to the low H₂O₂-tolerance of mitochondria, although the molecular mechanism of such a feedback inhibition is not well understood. In this study, the authors aimed to investigate the molecular mechanism of product inhibition of MnSOD through the structural characterization of a W161F variant of this enzyme, which was previously shown to be biased towards the product inhibited pathway. By solving the neutron structures of W161F MnSOD in different states and in the presence of D₂O₂, the authors showed that the Mn active site adopted a 5-coordinate Mn²⁺ form both in the resting state and upon peroxide treatment, an assignment further verified by HERFD-XAS spectroscopy. Moreover, they observed that an OOH moiety—proposed to be a hydroperoxyl anion (HO₂⁻) on the basis of XAS and DFT studies—was bound to Mn while forming an H-bond to Q143 in the D₂O₂-treated W161F variant, with a second H₂O₂ moiety at a distal site forming H-bonds with H30 and Y34. Notably, the authors revealed that the distal H₂O₂ moiety formed an extensive H-

bonding network that linked Q143 and the Mn active site, and that such an H-bonding network was absent in the resting state where the peroxide/hydroperoxyl anion binding sites were occupied by water molecules.

This is a well-conducted study that explores the mechanism of a significant process that regulates the activity of SOD and the intracellular level of H₂O₂. The authors used an excellent combination of structural, spectroscopic, and theoretical approaches to provide mutually supportive evidence in support of their conclusions; in particular, the XAS analysis that accompanied the neutron diffraction experiments was quality work that facilitated some of the key interpretations of the structural data, such as the assignment of a hydroperoxyl anion at the Mn active site. Overall, the quantity and quality of the data presented by the authors are outstanding, and the implications of their findings are more than intriguing with respect to both the inner workings of MnSODs and the PCET process in general. Because of these reasons, this work warrants publication in Nature Communications.

We thank the reviewer for their excitement about our work, and their feedback that has improved our manuscript!

There are, however, several key points that the authors should consider addressing prior to the publication of this manuscript:

(1) Based on their observation of a second, distal peroxide moiety, the authors suggest that product inhibition could be initiated by H₂O₂ that has already been formed instead of directly by the product that was on-path to its exit. This is a significant finding; however, it is unclear whether the extra peroxide is physiologically relevant or if it is an artifact resulting from the high concentration of H₂O₂ used in the experiments. The authors should consider including some relevant kinetic data to strengthen their argument, as product inhibition could very well occur at an exogenous H₂O₂ concentration that is within the range of the reactivity of MnSOD.

Along with a suggestion made by reviewer #2 – we have reframed our suggested mechanism where product inhibition is dependent on a molecule of O₂^{•-} arriving and reducing Mn³⁺ before the departure of a recently formed H₂O₂ (the distal peroxide moiety). This H₂O₂ hydrogen bonded between Tyr34 and His30 is not likely to be an artifact since the space in between these two residues is the only means for substrate/product to enter/exit the active site. Furthermore, His30 and Tyr34 engaging in proton transfers with a dioxygen species is plausible since these residues undergo changes in their protonation state without H₂O₂ soaking; please see our previous work¹. However, we now make several mentions in the text that high concentrations of D₂O₂/H₂O₂ were used to attain the data.

In regard to our experimental approach to attain the inhibited complex with high H₂O₂ concentrations – we now postulate that the inhibited complex we observe is from first a backward reaction of H₂O₂ with Mn³⁺SOD to form a O₂^{•-} species and Mn²⁺SOD, and then a forward reaction of O₂^{•-} with another Mn³⁺SOD in the presence of H₂O₂ to form the inhibited complex. Here, H₂O₂ is seen as a means to generate O₂^{•-} and study the forward reactions, and we have framed our proposed mechanism only in the forward direction to better represent what would occur in physiological conditions. These details have been added to our suggested mechanism in lines 645-649.

To support these conclusions, we now include details from Bull *et al.* that note mixing Mn³⁺SOD and H₂O₂ leads to a slow reaction yielding Mn²⁺SOD¹⁶. The process was best fit by second-order kinetics, where reduction of Mn³⁺SOD and H₂O₂ results in Mn²⁺SOD and O₂^{•-} (a backward reaction), and then a subsequent reaction of O₂^{•-} with a second Mn³⁺SOD to yield another Mn²⁺SOD site (a forward reaction). For a product-inhibited variant, like Trp161Phe, this forward reaction would instead be the formation of

the inhibited complex since the fast Mn^{3+} to Mn^{2+} redox reaction (k_2) is ablated. Like our work, several other studies have further observed that introducing high concentrations of H_2O_2 to Mn^{3+} SOD leads to the formation of the inhibited complex^{5,12,13,16}. These details have been added to the last paragraphs of the section “The electronic identity of the product-inhibited complex” in lines 378-397 and in the introduction in lines 155-163.

For our neutron diffraction and XAS experiments, 280 mM D_2O_2/H_2O_2 was needed to unambiguously isolate the product-inhibited Trp161Phe MnSOD complex in a crystal or 3 mM protein solution. Past studies correlate product-inhibition achieved through mixing H_2O_2 to the measured kinetics in the forward dismutation direction (i.e., reactions with $O_2^{\bullet-}$, **Table 1**)¹³. For example, variants with $k_2 \ll k_3$ (**Table 1**), like Trp161Phe, enrich the inhibited complex with excessive molar ratios of H_2O_2 . Indeed, mixing Mn^{3+} SOD with either $O_2^{\bullet-}$ or excessive concentrations of H_2O_2 leads to the same characteristic UV spectra of product inhibition (please see Fig. 1 in Hearn *et al.* 1999)¹³. These past observations suggest that product inhibition achieved through mixing H_2O_2 with Mn^{3+} SOD may be a combination of reverse and forward reactions. These details have now been added to the last paragraphs of the section “The electronic identity of the product-inhibited complex” in lines 386-397 and in the introduction in lines 155-163.

(2) It is unclear whether the state generated by soaking with H_2O_2 faithfully represents a naturally occurring inhibited state due to an accumulation of the product. Can this effect be verified by comparing the kinetic or spectroscopic features of the H_2O_2 treated sample with the substrate turnover sample? Would prolonged reactions and accumulation of products, like those occurring in the cell, lead to similar structures or states observed for the H_2O_2 soaked enzyme?

Mixing Mn^{3+} SOD with either $O_2^{\bullet-}$ or excessive concentrations of H_2O_2 leads to the same characteristic UV spectra of product inhibition¹³. This suggests that H_2O_2 soaking indeed reflects a product-inhibited complex attained from $O_2^{\bullet-}$. We have added these details to the manuscript in lines 395-396.

Indeed, high concentrations of H_2O_2 that we soak the enzyme in are not physiologically possible. Instead, we think of H_2O_2 as a means to generate $O_2^{\bullet-}$ and instigate the forward reaction that would occur in cells.

(3) It appears that the extensive hydrogen bonding network observed in the peroxide bound structure (Fig. 6a) could stabilize the intermediate-bound active site and thereby stall the reaction. This possibility could be tested by mutating H30 and Y34 or performing preliminary MD calculations that compare the energetics of the peroxide bound state (Fig. 6a) with those of the other states.

This is a wonderful suggestion! However, given the density of the data and analysis in our manuscript, we believe any further experiments would defocus the work – though these are compelling research directions we are currently pursuing. Obtaining the beamtime to perform neutron scattering or XAS is difficult given the high demand for these beamlines and long wait times.

A few minor points for the authors' consideration:

a. The background section, particular with respect to the kinetic features of product inhibition, does not provide adequate information for a layman. The paragraph from Ln 91 to Ln 100 should be expanded given the crucial importance of the phenomenon described herein for this study.

This is a good point. We have expanded the introduction to provide an improved description of the kinetics of MnSOD in lines 110-125.

b. In Figs 2 and 5, the authors jump between chain A and chain B across the different states (D₂O₂ soaked, reduced, and oxidized states) of the enzyme. For the purpose of a better comparison, the authors should include a parallel illustration of chain A and B for all 3 reported states in the supplemental information.

This is a great suggestion and parallel illustrations are now shown in Supplementary Fig. 2.

c. Fig. 6 is a great figure summarizing all the findings in this manuscript. However, it would be helpful for the readers if the authors could cross reference the structures shown in this figure with the various experiments described in the manuscript.

Another great suggestion! In the Fig 6 legend, we now cross-reference the structures with corresponding experimental data. Additionally, we reference Supplementary Fig. 2, which show parallel illustrations of all neutron active sites.

Reviewer #4 (Remarks to the Author):

Azadmanesh and co-authors have determined neutron structures of human manganese superoxide dismutase (MnSOD) with/without D₂O₂ soaking. Neutron structures elucidated the protonation states of amino acids and the hydrogen bond network in the D₂O₂-soaked, reduced, and oxidized MnSOD W161F mutant. The comparison between these structures, including MnSOD WT structures, allows us to understand the PCET catalysis of MnSOD. In addition, XAS and QM (DFT) calculations correlated well with neutron structures and complemented them. Therefore, the proposed atomic and electronic mechanism for MnSOD product inhibition and relief should be supported. On the other hand, the hydrogen bond information, especially SSHB, is unreliable because the coordinates of many hydrogens (deuteriums) are shifted from the HD-omit neutron maps. The authors should mention these deviations and, if possible, provide coordinates and structure factors to the reviewers.

We kindly thank the reviewer for the supportive review of our manuscript and for providing helpful feedback. In regard to the D atoms being shifted from the corresponding omit maps – the presence or absence of D is the more compelling information at 2.3 Å resolution, and small offsets between density and D positions are expected at this resolution. These offsets are also observed for other published neutron structures of comparable resolutions^{1,2,23}. Lowering the sigma levels of the omit maps also cover the D atoms. We have also incorporated coordinate error statistics into Supplementary Table 5 and note that the disorder of atom positions is also reflected by the *B*-factor.

While the hydrogen bonding information is subject to experimental uncertainty and the limits of our resolution, we are careful during our model building and refinement to place the most chemically sensible arrangement of atoms from the density given – and report the resulting hydrogen bond distances. If we were to ensure the D atom positions are centered within omit density centers, unrealistic O-D and N-D covalent bond lengths would result and defy chemical reasonability. To address this issue, we have reiterated the resolution of our neutron diffraction data in the accompanying figures legends – of both the main manuscript and supplement. Furthermore, we now address the deviations between density and D positions from our resolution in the methods section “Crystallographic Data Processing and Refinement” in lines 776-778.

Other points

1. As the authors mentioned, neutron crystallography gives us structure information without radiation-induced artifacts. Do the authors consider the X-ray structure information shown in supplementary Fig. 1 to be an artifact? If so, the authors should exclude X-ray structure analyses from the manuscript. (The supplementary Fig. 1 will only confuse the readers).

Yes, we do consider the X-ray structures shown in Supplementary Fig. 1 influenced by radiation-induced artifacts. We see the reviewer's point that it may indeed confuse the readers – though we feel including Supplementary Fig. 1 strengthens the rationale for pursuing neutron crystallography and bolsters the overall impact of the work. We have described both in the main manuscript and the Supplementary Figure 1 legend that the X-ray structures are influenced by photoreduction effects. We also altered the sentence following the first mention of Supplementary Fig. 1 to note the dioxygen molecules in these X-ray structures are different in orientation and partially occupied due to X-ray irradiation – and demonstrate the benefit of using neutrons for photoreduction-free structural analysis. Additionally, we added the occupancy numbers to Supplementary Fig. 1 to accentuate the concept.

2. The authors used W161F mutant to trap the product-inhibited complex. Have they tried using wild type? By controlling the soaking times, the oxidized resting state and inhibitor complex may be distinguished. If so, this manuscript will be even more substantial.

Yes, this strategy is extremely appealing; however, wildtype does not enrich the product-inhibited complex with an occupancy as high as W161F, as this would potentially be physiologically deleterious to cells. The XAS spectrum of product-inhibited wildtype (Supplementary Fig. 3a) closely resembles the spectrum of reduced wildtype. This indicates that the measured sample is a mixture of species between the inhibited complex and the reduced complex. Trp161Phe, on the other hand, shows distinct spectra for peroxide treated, indicating that the product-inhibited form is not a mixture. While we mention this in the main text, we added details to the legend of Supplementary Fig. 3 in an effort to make this clearer.

3. The authors should add bond lengths from DFT calculation using a hydroperoxy radical (HO_2^\cdot) to Table 2.

The corresponding bond lengths have been added to Table 2.

4. The authors should show unoccupied residual density by calculating D_2O , not D_2O_2 , as a supplementary figure.

As suggested, we have added a new Supplementary Figure 5 to show the residual density from modeling D_2O .

Minor points

1. line 501: Fig. 3a → Fig. 5a2.

Thank you, we have corrected the text.

2. Which crystals were used for neutron structure analyses, obtained in space or on the ground?

Good point! The D_2O_2 -soaked and reduced Trp161Phe MnSOD neutron data sets were collected from crystals grown in microgravity, while all other data sets were grown from crystals grown on

Earth. These details have been added to the “Crystallographic Data Collection” methods section, and a line for “crystal growth environment” has been added to Supplementary Table 5.

3. The $CC_{1/2}$ values in the outer shell of W161F reduced and oxidized neutron structures below 0.5. Are these d_{\min} adequate?

Given the high multiplicity (>3.7), generous $I/\sigma(I)$ values (>3.0), and good completeness (>87%) at these high-resolution shells, we believe the d_{\min} values are appropriate. There are several works (summarized by Karplus and Diederichs²⁴) that support including data corresponding to $CC_{1/2}$ values above 0.2. In brief, the inclusion of these reflections has a positive impact on refinement and map quality.

References Cited

- 1 Azadmanesh, J., Lutz, W. E., Coates, L., Weiss, K. L. & Borgstahl, G. E. O. Direct detection of coupled proton and electron transfers in human manganese superoxide dismutase. *Nat Commun* **12**, 2079 (2021). <https://doi.org/10.1038/s41467-021-22290-1>
- 2 Dajnowicz, S. et al. Direct visualization of critical hydrogen atoms in a pyridoxal 5'-phosphate enzyme. *Nat Commun* **8**, 955 (2017). <https://doi.org/10.1038/s41467-017-01060-y>
- 3 Oksanen, E., Chen, J. C. & Fisher, S. Z. Neutron Crystallography for the Study of Hydrogen Bonds in Macromolecules. *Molecules* **22** (2017). <https://doi.org/10.3390/molecules22040596>
- 4 Gajdos, L. et al. Neutron crystallography reveals mechanisms used by *Pseudomonas aeruginosa* for host-cell binding. *Nat Commun* **13**, 194 (2022). <https://doi.org/10.1038/s41467-021-27871-8>
- 5 Hearn, A. S. et al. Kinetic analysis of product inhibition in human manganese superoxide dismutase. *Biochemistry* **40**, 12051-12058 (2001).
- 6 Garman, E. F. & Weik, M. Radiation Damage in Macromolecular Crystallography. *Methods Mol. Biol.* **1607**, 467-489 (2017). https://doi.org/10.1007/978-1-4939-7000-1_20
- 7 Garman, E. F. & Weik, M. Macromolecular crystallography radiation damage research: what's new? *J. Synchrotron. Radiat.* **18**, 313-317 (2011). <https://doi.org/10.1107/S0909049511013859>
- 8 Garman, E. F. & Owen, R. L. Cryocooling and radiation damage in macromolecular crystallography. *Acta Cryst. D* **62**, 32-47 (2006). <https://doi.org/10.1107/S09074444905034207>
- 9 Azadmanesh, J., Lutz, W. E., Coates, L., Weiss, K. L. & Borgstahl, G. E. O. Cryotrapping peroxide in the active site of human mitochondrial manganese superoxide dismutase crystals for neutron diffraction. *Acta Cryst. F* **78**, 8-16 (2022). <https://doi.org/10.1107/S2053230X21012413>
- 10 Abreu, I. A. & Cabelli, D. E. Superoxide dismutases—a review of the metal-associated mechanistic variations. *Biochim Biophys Acta* **1804**, 263-274 (2010). <https://doi.org/10.1016/j.bbapap.2009.11.005>
- 11 Sheng, Y. et al. Superoxide dismutases and superoxide reductases. *Chem. Rev.* **114**, 3854-3918 (2014). <https://doi.org/10.1021/cr4005296>
- 12 Cabelli, D. E. et al. Role of tryptophan 161 in catalysis by human manganese superoxide dismutase. *Biochemistry* **38**, 11686-11692 (1999).
- 13 Hearn, A. S., Tu, C., Nick, H. S. & Silverman, D. N. Characterization of the product-inhibited complex in catalysis by human manganese superoxide dismutase. *J. Biol. Chem.* **274**, 24457-24460 (1999).
- 14 Srncic, M., Aquilante, F., Ryde, U. & Rulisek, L. Reaction Mechanism of Manganese Superoxide Dismutase Studied by Combined Quantum and Molecular Mechanical Calculations and Multiconfigurational Methods. *J. Phys. Chem. B*, 6074-6086 (2009).
- 15 Leveque, V. J., Vance, C. K., Nick, H. S. & Silverman, D. N. Redox properties of human manganese superoxide dismutase and active-site mutants. *Biochemistry* **40**, 10586-10591 (2001).
- 16 Bull, C., Niederhoffer, E. C., Yoshida, T. & Fee, J. A. Kinetic Studies of Superoxide Dismutases: Properties of the Manganese-Containing Protein from *Thermus thermophilus*. *J. Am. Chem. Soc.* **113**, 4069-4076 (1991).
- 17 Bai, J. & Cederbaum, A. I. Mitochondrial catalase and oxidative injury. *Biol. Signals Recept.* **10**, 189-199 (2001). <https://doi.org/10.1159/000046887>

- 18 Palma, F. R. *et al.* Mitochondrial Superoxide Dismutase: What the Established, the Intriguing, and the Novel Reveal About a Key Cellular Redox Switch. *Antioxid. Redox Signal.* **32**, 701-714 (2020).
<https://doi.org:10.1089/ars.2019.7962>
- 19 Riemer, J., Schwarzlander, M., Conrad, M. & Herrmann, J. M. Thiol switches in mitochondria: operation and physiological relevance. *Biol. Chem.* **396**, 465-482 (2015). <https://doi.org:10.1515/hsz-2014-0293>
- 20 Davis, C. A. *et al.* Potent anti-tumor effects of an active site mutant of human manganese-superoxide dismutase. Evolutionary conservation of product inhibition. *J. Biol. Chem.* **279**, 12769-12776 (2004).
<https://doi.org:10.1074/jbc.M310623200>
- 21 Bunau, O. & Joly, Y. Self-consistent aspects of x-ray absorption calculations. *J Phys Condens Matter* **21**, 345501 (2009). <https://doi.org:10.1088/0953-8984/21/34/345501>
- 22 Sheng, Y. *et al.* Comparison of two yeast MnSODs: mitochondrial *Saccharomyces cerevisiae* versus cytosolic *Candida albicans*. *J. Am. Chem. Soc.* **133**, 20878-20889 (2011).
<https://doi.org:10.1021/ja2077476>
- 23 Schroder, G. C., O'Dell, W. B., Webb, S. P., Agarwal, P. K. & Meilleur, F. Capture of activated dioxygen intermediates at the copper-active site of a lytic polysaccharide monooxygenase. *Chem. Sci.* **13**, 13303-13320 (2022). <https://doi.org:10.1039/d2sc05031e>
- 24 Karplus, P. A. & Diederichs, K. Assessing and maximizing data quality in macromolecular crystallography. *Curr. Opin. Struct. Biol.* **34**, 60-68 (2015). <https://doi.org:10.1016/j.sbi.2015.07.003>

REVIEWERS' COMMENTS

Reviewer #1 (Remarks to the Author):

This is a revised manuscript, presenting three neutron structures of the W161F mutant of the MnSOD, viz. the reduced and oxidised resting states and the peroxide-inhibited structure. The authors have satisfactorily answered my previous question and modified the manuscript accordingly. It can be published as it is now.

Reviewer #2 (Remarks to the Author):

This is a very exciting piece of work. I am sorry that the authors never received the file in which I made so many suggestions, but understand that this is not their fault. The authors have responded in detail to the comments of the other reviewers, and addressed their concerns.

Reviewer #3 (Remarks to the Author):

The authors have made great efforts to address my previous concerns, particularly with respect to the physiological relevance of the high H₂O₂ concentrations used to generate the structure of the inhibited enzyme. Not only have the authors revised their mechanistic proposal to highlight the relevance of this structure, but they have also recapitulated how H₂O₂ could become a means to generate the experimentally observed superoxide via a backward step that is suggested by previous kinetic data. As such, my concerns have been thoroughly addressed, and the revised manuscript is ready for publication.

Reviewer #4 (Remarks to the Author):

The authors generally corresponded to all referee's comments. The revised manuscript is suitable for publication in Nature Communications.